# Outlier-Robust High-Dimensional Sparse Estimation via Iterative Filtering

**Ilias Diakonikolas**
University of Wisconsin - Madison
ilias.diakonikolas@gmail.com

**Sushrut Karmalkar**
UT Austin
s.sushrut@gmail.com

**Daniel Kane**
University of California, San Diego
dakane@ucsd.edu

**Eric Price**
UT Austin
ecprice@cs.utexas.edu

**Alistair Stewart**
Web3 Foundation
stewart.al@gmail.com

## Abstract

We study high-dimensional sparse estimation tasks in a robust setting where a constant fraction of the dataset is adversarially corrupted. Specifically, we focus on the fundamental problems of robust sparse mean estimation and robust sparse PCA. We give the first practically viable robust estimators for these problems. In more detail, our algorithms are sample and computationally efficient and achieve near-optimal robustness guarantees. In contrast to prior provable algorithms which relied on the ellipsoid method, our algorithms use spectral techniques to iteratively remove outliers from the dataset. Our experimental evaluation on synthetic data shows that our algorithms are scalable and significantly outperform a range of previous approaches, nearly matching the best error rate without corruptions.

## 1 Introduction

### 1.1 Background

The task of leveraging sparsity to extract meaningful information from high-dimensional datasets is a fundamental problem of significant practical importance, motivated by a range of data analysis applications. Various formalizations of this general problem have been investigated in statistics and machine learning for at least the past two decades, see, e.g., [HTW15] for a recent textbook on the topic. This paper focuses on the *unsupervised setting* and in particular on estimating the parameters of a high-dimensional distribution under sparsity assumptions. Concretely, we study the problems of *sparse mean estimation* and *sparse PCA* under natural data generating models.

The classical setup in statistics is that the data was generated by a probabilistic model of a given type. This is a simplifying assumption that is only approximately valid, as real datasets are typically exposed to some source of contamination. The field of robust statistics [Hub64, HR09, HRRS86] aims to design estimators that are *robust* in the presence of model misspecification. In recent years, designing computationally efficient robust estimators for high-dimensional settings has become a pressing challenge in a number of applications. These include the analysis of biological datasets, where natural outliers are common [RPW+02, PLJD10, LAT+08] and can contaminate the downstream statistical analysis, and *data poisoning attacks* [BNJT10], where even a small fraction of fake data (outliers) can substantially degrade the learned model [BNL12, SKL17].

This discussion motivates the design of robust estimators that can tolerate a *constant* fraction of adversarially corrupted data. We will use the following model of corruptions (see, e.g., [DKK+16]):

**Definition 1.1.** *Given $0 < \varepsilon < 1/2$ and a family of distributions $\mathcal{D}$ on $\mathbb{R}^d$, the* adversary *operates as follows: The algorithm specifies some number of samples $N$, and $N$ samples $X_1, X_2, \ldots, X_N$ are drawn from some (unknown) $D \in \mathcal{D}$. The adversary is allowed to inspect the samples, removes $\varepsilon N$ of them, and replaces them with arbitrary points. This set of $N$ points is then given to the algorithm. We say that a set of samples is $\varepsilon$-corrupted if it is generated by the above process.*

Our model of corruptions generalizes several other robustness models, including Huber's contamination model [Hub64] and the malicious PAC model [Val85, KL93].

In the context of robust sparse mean estimation, we are given an $\varepsilon$-corrupted set of samples from an unknown mean Gaussian distribution $\mathcal{N}(\mu, I)$, where $\mu \in \mathbb{R}^d$ is assumed to be $k$-sparse, and the goal is to output a hypothesis vector $\widehat{\mu}$ that approximates $\mu$ in $\ell_2$-norm. In the context of robust sparse PCA (in the spiked covariance model), we are given an $\varepsilon$-corrupted set of samples from $\mathcal{N}(\mathbf{0}, \rho v v^T)$, where $v \in \mathbb{R}^d$ is assumed to be $k$-sparse and the goal is to approximate $v$. In both settings, we would like to design computationally efficient estimators with sample complexity $\text{poly}(k, \log d, 1/\varepsilon)$, i.e., close to the information theoretic minimum, that achieve near-optimal error guarantees.

Until recently, even for the simplest high-dimensional parameter estimation settings, no polynomial time robust learning algorithms with dimension-independent error guarantees were known. Two concurrent works [DKK$^+$16, LRV16] made the first progress on this front for the unsupervised setting. Specifically, [DKK$^+$16, LRV16] gave the first polynomial time algorithms for robustly learning the mean and covariance of high-dimensional Gaussians and other models. These works focused on the dense regime and as a result did not obtain algorithms with sublinear sample complexity in the sparse setting. Building on [DKK$^+$16], more recent work [BDLS17] obtained sample efficient polynomial time algorithms for the robust *sparse* setting, and in particular for the problems of robust sparse mean estimation and robust sparse PCA studied in this paper. These algorithms are based the unknown convex programming methodology of [DKK$^+$16] and in particular *inherently rely on the ellipsoid algorithm*. Moreover, the separation oracle required for the ellipsoid algorithm turns out to be another convex program — corresponding to an SDP to solve sparse PCA. As a consequence, the running time of these algorithms, while polynomially bounded, is impractically high.

## 1.2 Our Results and Techniques

The main contribution of this paper is the design of significantly faster robust estimators for the aforementioned high-dimensional sparse problems. More specifically, our algorithms are iterative and each iteration involves a simple spectral operation (computing the largest eigenvalue of an approximate matrix). Our algorithms achieve the same error guarantee as [BDLS17] with similar sample complexity. At the technical level, we enhance the *iterative filtering methodology* of [DKK$^+$16] to the sparse setting, which we believe is of independent interest and could lead to faster algorithms for other robust sparse estimation tasks as well.

For robust sparse mean estimation, we show:

**Theorem 1.2** (Robust Sparse Mean Estimation). *Let $D \sim \mathcal{N}(\mu, I)$ be a Gaussian distribution on $\mathbb{R}^d$ with unknown $k$-sparse mean vector $\mu$, and $\varepsilon > 0$. Let $S$ be an $\varepsilon$-corrupted set of samples from $D$ of size $N = \widetilde{\Omega}(k^2 \log(d)/\varepsilon^2)$. There exists an algorithm that, on input $S$, $k$, and $\varepsilon$ runs in polynomial time returns $\widehat{\mu}$ such that with probability at least $2/3$ it holds $\|\widehat{\mu} - \mu\|_2 = O(\varepsilon \sqrt{\log(1/\varepsilon)})$.*

Some comments are in order. First, the sample complexity of our algorithm is asymptotically the same as that of [BDLS17], and matches the lower bound of [DKS17] against Statistical Query algorithms for this problem. The major advantage of our algorithm over [BDLS17] is that while their algorithm made use of the ellipsoid method, ours uses only spectral techniques and is scalable.

For robust sparse PCA in the spiked covariance model, we show:

**Theorem 1.3** (Robust Sparse PCA). *Let $D \sim \mathcal{N}(\mathbf{0}, I + \rho v v^T)$ be a Gaussian distribution on $\mathbb{R}^d$ with spiked covariance for an unknown $k$-sparse unit vector $v$, and $0 < \rho < O(1)$. For $\varepsilon > 0$, let $S$ be an $\varepsilon$-corrupted set of samples from $D$ of size $N = \Omega(k^4 \log^4(d/\varepsilon)/\varepsilon^2)$. There exists an algorithm that, on input $S$, $k$, and $\varepsilon$, runs in polynomial time and returns $\hat{v} \in \mathbb{R}^d$ such that with probability at least $2/3$ we have that $\|\hat{v}\hat{v}^T - vv^T\|_F = O\left(\varepsilon \log(1/\varepsilon)/\rho\right)$.*

The sample complexity upper bound in Theorem 1.3 is somewhat worse than the information theoretic optimum of $\Theta(k^2 \log d / \varepsilon^2)$. While the ellipsoid-based algorithm of [BDLS17] achieves near-optimal sample complexity (within logarithmic factors), our algorithm is practically viable as it only uses spectral operations. We also note that the sample complexity in our above theorem is not known to be optimal for our algorithm. It seems quite plausible, via a tighter analysis, that our algorithm in fact has near-optimal sample complexity as well.

For both of our algorithms, in the most interesting regime of $k \ll \sqrt{d}$, the running time per iteration is dominated by the $O(Nd^2)$ computation of the empirical covariance matrix. The number of iterations is at most $\varepsilon N$, although it typically is much smaller, so both algorithms take at most $O(\varepsilon N^2 d^2)$ time.

## 1.3 Related Work

There is extensive literature on exploiting sparsity in statistical estimation (see, e.g., [HTW15]). In this section, we summarize the related work that is directly related to the results of this paper. Sparse mean estimation is arguably one of the most fundamental sparse estimation tasks and is closely related to the Gaussian sequence model [Tsy08, Joh17]. The task of sparse PCA in the spiked covariance model, initiated in [Joh01], has been extensively investigated (see Chapter 8 of [HTW15] and references therein). In this work, we design algorithms for the aforementioned problems that are robust to a constant fraction of outliers.

Learning in the presence of outliers is an important goal in statistics studied since the 1960s [Hub64]. See, e.g., [HR09, HRRS86] for book-length introductions in robust statistics. Until recently, all known computationally efficient high-dimensional estimators could tolerate a negligible fraction of outliers, even for the task of mean estimation. Recent work [DKK+16, LRV16] gave the first efficient robust estimators for basic high-dimensional unsupervised tasks, including mean and covariance estimation. Since the dissemination of [DKK+16, LRV16], there has been a flurry of research activity on computationally efficient robust learning in high dimensions [BDLS17, CSV17, DKK+17, DKS17, DKK+18a, SCV18, DKS18b, DKS18a, HL18, KSS18, PSBR18, DKK+18b, KKM18, DKS19, LSLC18a, CDKS18, CDG18, CDGW19].

In the context of robust sparse estimation, [BDLS17] obtained sample-efficient and polynomial time algorithms for robust sparse mean estimation and robust sparse PCA. The main difference between [BDLS17] and the results of this paper is that the [BDLS17] algorithms use the ellipsoid method (whose separation oracle is an SDP). Hence, these algorithms are prohibitively slow for practical applications. More recent work [LSLC18b] gave an iterative method for robust sparse mean estimation, which however requires multiple solutions to a convex relaxation for sparse PCA in each iteration. Finally, [LLC19] proposed an algorithm for robust sparse mean estimation via iterative trimmed hard thresholding. While this algorithm seems practically viable in terms of runtime, it can only tolerate $1/(\sqrt{k} \log(nd))$ – i.e., *sub-constant* – fraction of corruptions.

## 1.4 Paper Organization

In Section 2, we describe our algorithms and provide a detailed sketch of their analysis. In Section 3, we report detailed experiments demonstrating the performance of our algorithms on synthetic data in various parameter regimes. Due to space limitations, the full proofs of correctness for our algorithms can be found in the full version of this paper.

## 2 Algorithms

In this section, we describe our algorithms in tandem with a detailed outline of the intuition behind them and a sketch of their analysis. Due to space limitations, the proof of correctness is deferred to the full version of our paper.

At a high-level, our algorithms use the iterative filtering methodology of [DKK+16]. The main idea is to iteratively remove a small subset of the dataset, so that eventually we have removed all the important outliers and the standard estimator (i.e., the estimator we would have used in the noiseless case) works. Before we explain our new ideas that enhance the filtering methodology to the sparse setting, we provide a brief technical description of the approach.

**Overview of Iterative Filtering.** The basic idea of iterative filtering [DKK$^+$16] is the following: In a given iteration, carefully pick some test statistic (such as $v \cdot x$ for a well-chosen $v$). If there were no outliers, this statistic would follow a nice distribution (with good concentration properties). This allows us to do some sort of statistical hypothesis testing of the "null hypothesis" that each $x_i$ is an inlier, rejecting it (and believing that $x_i$ is an outlier) if $v \cdot x_i$ is far from the expected distribution. Because there are a large number of such hypotheses, one uses a procedure reminiscent of the Benjamini-Hochberg procedure [BH95] to find a candidate set of outliers with low *false discovery rate* (FDR), i.e., a set with more outliers than inliers in expectation. This procedure looks for a threshold $T$ such that the fraction of points with test statistic above $T$ is at least a constant factor more than it "should" be. If such a threshold is found, those points are mostly outliers and can be safely removed. The key goal is to judiciously design a test statistic such that either the outliers aren't particularly important—so the naive empirical solution is adequate—or at least one point will be filtered out.

In other words, the goal is to find a test statistic such that, if the distribution of the test statistic is "close" to what it would be in the outlier-free world, then the outliers cannot perturb the answer too much. An additional complication is that the test statistics depend on the data (such as $v \cdot x$, where $v$ is the principal component of the data) making the distribution on inliers also nontrivial. This consideration drives the sample complexity of the algorithms.

In the algorithms we describe below, we use a specific parameterized notion of a good set. We define these precisely in the supplementary material, briefly, any large enough sample drawn from the uncorrupted distribution will satisfy the structural properties required for the set to be good.

We now describe how to design such test statistics for our two sparse settings.

**Notation** Before we describe our algorithms, we set up some notation. We define $h_k : \mathbb{R}^d \to \mathbb{R}^d$ to be the thresholding operator that keeps the $k$ entries of $v$ with the largest magnitude and sets the rest to 0. For a finite set $S$, we will use $a \in_u S$ to mean that $a$ is chosen uniformly at random from $S$. For $M \in \mathbb{R}^d \times \mathbb{R}^d$ and $U \subseteq [d]$, let $M_U$ denote the matrix $M$ restricted to the $U \times U$ submatrix.

---

**Algorithm 1** Robust Sparse Mean Estimation via Iterative Filtering

1: **procedure** ROBUST-SPARSE-MEAN$(S, k, \varepsilon, \tau)$

**input:** A multiset $S$ such that there exists an $(\varepsilon, k, \tau)$-good set $G$ with $\Delta(G, S) \leq 2\varepsilon$.

**output:** Multiset $S'$ with smaller fraction of corrupted samples or a vector $\widehat{\mu}$ with $\|\widehat{\mu} - \mu\|_2 \leq \varepsilon\sqrt{\log(1/\varepsilon)}$.

2:     Compute the sample mean $\widetilde{\mu} = \mathbf{E}_{X \in_u S}[X]$ and the sample covariance matrix $\widetilde{\Sigma}$, i.e., $\widetilde{\Sigma} = (\widetilde{\Sigma}_{i,j})_{1 \leq i,j \leq d}$ with $\widetilde{\Sigma}_{i,j} = \mathbf{E}_{X \in_u S}[(X_i - \widetilde{\mu}_i)(X_j - \widetilde{\mu}_j)]$.

3:     Let $U \subseteq [d] \times [d]$ be the set of the $k$ largest magnitude entries of the diagonal of $\widetilde{\Sigma} - I$ and the largest magnitude $k^2 - k$ off-diagonal entries, with ties broken so that if $(i, j) \in U$ then $(j, i) \in U$.

4:     **if** $\|(\widetilde{\Sigma} - I)_{(U)}\|_F \leq O(\varepsilon \log(1/\varepsilon))$ **then return** $\widehat{\mu} := h_k(\widetilde{\mu})$.

5:     Set $U' = \{i \in [d] : (i, j) \in U\}$.

6:     Compute the largest eigenvalue $\lambda^*$ of $(\widetilde{\Sigma} - I)_{U'}$ and a corresponding unit eigenvector $v^*$.

7:     **if** $\lambda^* \geq \Omega(\varepsilon\sqrt{\log(1/\varepsilon)})$ **then**: Let $\delta_\ell := 3\sqrt{\varepsilon\lambda^*}$. Find $T > 0$ such that

$$\mathbf{Pr}_{X \in_u S}\left[|v^* \cdot (X - \widetilde{\mu})| \geq T + \delta_\ell\right] \geq 9 \cdot \mathrm{erfc}(T/\sqrt{2}) + \frac{3\varepsilon^2}{T^2 \ln(k \ln(Nd/\tau))}.$$

8:         **return** the multiset $S' = \{x \in S : |v^* \cdot (x - \widetilde{\mu})| \leq T + \delta_\ell\}$.

9:     Let $p(x) = \left((x - \widetilde{\mu})^T (\widetilde{\Sigma} - I)_{(U)}^T (x - \widetilde{\mu}) - \mathrm{Tr}((\widetilde{\Sigma} - I)_{(U)})\right) / \|(\widetilde{\Sigma} - I)_{(U)}\|_F$.

10:    Find $T > 4$ such that

$$\mathbf{Pr}_{X \in_u S}[|p(X)| \geq T] \geq 9 \exp(-T/4) + 3\varepsilon^2/(T \ln^2 T).$$

11:    **return** the multiset $S' = \{x \in S : |p(x)| \leq T\}$.

---

**Robust Sparse Mean Estimation.** Here we briefly describe the motivation and analysis of Algorithm 1, describing a single iteration of our filter for the robust sparse mean setting.

In order to estimate the $k$-sparse mean $\mu$, it suffices to ensure that our estimate $\mu'$ has $|v \cdot (\mu' - \mu)|$ small for any $2k$-sparse unit vector $v$. The now-standard idea in robust statistics [DKK$^+$16] is that if a small number of corrupted samples suffice to cause a large change in our estimate of $v \cdot \mu$, then this must lead to a substantial increase in the sample variance of $v \cdot x$, which we can detect.

Thus, a very basic form of a robust algorithm might be to compute a sample covariance matrix $\widetilde{\Sigma}$, and let $v$ be the $2k$-sparse unit vector that maximizes $v^T \widetilde{\Sigma} v$. If this number is close to 1, it certifies that our estimate $\mu'$ — obtained by truncating the sample mean to its $k$-largest entries — is a good estimate of the true mean $\mu$. If not, this will allow us to filter our sample set by throwing away the values where $v \cdot x$ is furthest from the true mean. This procedure guarantees that we have removed more corrupted samples than uncorrupted ones. We then repeat the filter until the empirical variance in every sparse direction is close to 1.

Unfortunately, the optimization problem of finding the optimal $v$ is computationally challenging, requiring a convex program. To circumvent the need for a convex program, we notice that $v^T \widetilde{\Sigma} v - 1 = (\widetilde{\Sigma} - I) \cdot (vv^T)$ is large only if $\widetilde{\Sigma} - I$ has large entries on the $(2k)^2$ non-zero entries of $vv^T$. Thus, if the $4k^2$ largest entries of $\widetilde{\Sigma} - I$ had small $\ell_2$-norm, this would certify that no such bad $v$ existed and would allow us to return the truncated sample mean. In case these entries have large $\ell_2$-norm, we show that we can produce a filter that removes more bad samples than good ones. Let $A$ be the matrix consisting of the large entries of $\widetilde{\Sigma}$ (for the moment assume that they are all off diagonal, but this is not needed). We know that the sample mean of $p(x) = (x - \mu')^T A(x - \mu') = \widetilde{\Sigma} \cdot A = \|A\|_F^2$. On the other hand, if $\mu'$ approximates $\mu$ on the $O(k^2)$ entries in question, we would have that $\|p\|_2 = \|A\|_F$. This means that if $\|A\|_F$ is reasonably large, an $\varepsilon$-fraction of corrupted points changed the mean of $p$ from 0 to $\|A\|_F^2 = \|A\|_F \|p\|_2$. This means that many of these errors must have had $|p(x)| \ll \|A\|_F / \varepsilon \|p\|_2$. This becomes very unlikely for good samples if $\|A\|_F$ is much larger than $\varepsilon$ (by standard results on the concentration of Gaussian polynomials). Thus, if $\mu'$ is approximately $\mu$ on these $O(k^2)$ coordinates, we can produce a filter. To ensure this, we can use existing filter-based algorithms to approximate the mean on these $O(k^2)$ coordinates. This results in Algorithm 1. For the analysis, we note that if the entries of $A$ are small, then $v^T(\widetilde{\Sigma} - I)v$ must be small for any unit $k$-sparse $v$, which certifies that the truncated sample mean is good. Otherwise, we can filter the samples using the first kind of filter. This ensures that our mean estimate is sufficiently close to the true mean that we can then filter using the second kind of filter.

It is not hard to show that the above works if we are given sufficiently many samples, but to obtain a tight analysis of the sample complexity, we need a number of subtle technical ideas. The detailed analysis of the sample complexity is deferred to the full version of our paper.

**Robust Sparse PCA** Here we briefly describe the motivation and analysis of Algorithm 2, describing a single iteration of our filter for the sparse PCA setting.

Note that estimating the $k$-sparse vector $v$ is equivalent to estimating $\mathbf{E}[XX^T - I] = vv^T$. In fact, estimating $\mathbf{E}[XX^T - I]$ to error $\varepsilon$ in Frobenius norm allows one to estimate $v$ within error $\varepsilon$ in $\ell_2$-norm. Thus, we focus on he task of robustly approximating the mean of $Y = XX^T - I$.

Our algorithm is going to take advantage of one fact about $X$ that even errors cannot hide: that $\mathbf{Var}[v \cdot X]$ is large. This is because removing uncorrupted samples cannot reduce the variance by much more than an $\varepsilon$-fraction, and adding samples can only increase it. This means that an adversary attempting to fool our algorithm can only do so by creating other directions where the variance is large, or simply by adding other large entries to the sample covariance matrix in order to make it hard to find this particular $k$-sparse eigenvector. In either case, the adversary is creating large entries in the empirical mean of $Y$ that should not be there. This suggests that the largest entries of the empirical mean of $Y$, whether errors or not, will be of great importance.

These large entries will tell us where to focus our attention. In particular, we can find the $k^2$ largest entries of the empirical mean of $Y$ and attempt to filter based on them. When we do so, one of two things will happen: Either we remove bad samples and make progress or we verify that these entries ought to be large, and thus must come from the support of $v$. In particular, when we reach the second

---

**Algorithm 2** Robust Sparse PCA via Iterative Filtering

---

1: **procedure** ROBUST-SPARSE-PCA$(S, k, \widetilde{\Sigma}, \varepsilon, \delta, \tau)$

**input:** A multiset S, an estimate of the true covariance $\widetilde{\Sigma}$, a real number $\delta \in \mathbb{R}$.

**output:** A multiset $S'$ with smaller fraction of corrupted samples or a matrix $\Sigma'$ with $\|\Sigma' - \Sigma\|_F \leq$
$\quad O(\sqrt{\varepsilon\delta} + \varepsilon \log(1/\varepsilon))$

2: $\quad$ For any $x \in \mathbb{R}^d$ define $\gamma(x) := \mathsf{vec}(xx^T - I) \in \mathbb{R}^{d^2}$.

3: $\quad$ Compute $\tilde{\mu} := \mathbf{E}_S[\gamma(x)]$, $\hat{\mu} = h_{k^2}(\mu)$ and $Q := \mathsf{Supp}(\hat{\mu})$.

4: $\quad$ Compute

$$M_Q := \mathbf{E}_S[(\gamma(x) - \tilde{\mu})(\gamma(x) - \tilde{\mu})^T]_{Q \times Q} \in \mathbb{R}^{k^2} \times \mathbb{R}^{k^2}$$

5: $\quad$ Let $\lambda, v^*$ be the maximum eigenvalue and corresponding eigenvector of $M_Q -$
$\quad$ $\mathrm{Cov}_{X \sim \mathcal{N}(0,\widetilde{\Sigma})}(\gamma(x)_Q)$.

6: $\quad$ **if** $\lambda < C \cdot (\delta + \varepsilon \log^2(1/\varepsilon))$, where $C$ is a sufficiently large constant **then**

7: $\quad\quad$ Compute $w$, the largest eigenvector of $\mathsf{mat}(\tilde{\mu})_Q$. **return** $ww^T + I$.

8: $\quad$ Let $\hat{\mu} = \mathsf{median}\left(\{\gamma(x) \cdot v^* \mid x \in S\}\right)$. Find a number $T > \log(1/\varepsilon)$ satisfying

$$\mathbf{Pr}_S[|\gamma(x)_Q \cdot v^* - \hat{\mu}| > CT + 3] > \frac{\varepsilon}{T^2 \log^2(T)}.$$

$\quad$ **return** $S' = \{x \in S \mid |(\gamma(x)_Q \cdot v^*) - \hat{\mu}| < T\}$.

---

case, since the adversary cannot shrink the empirical variance of $v \cdot X$ by much, almost all of the entries on the support of $v$ must remain large, and this can be captured by our algorithm.

The above algorithm works under a set of deterministic conditions on the good set of samples that are satisfied with high probability with $\mathrm{poly}(k) \log(d)/\varepsilon^2$ samples. Our current analysis does not establish the information-theoretically optimal sample size of $O(k^2 \log(d)/\varepsilon^2)$, though we believe that this plausible via a tighter analysis.

We note that a naive implementation of this algorithm will achieve error $\mathrm{poly}(\varepsilon)$ in our final estimate for $v$, while our goal is to obtain $\tilde{O}(\varepsilon)$ error. To achieve this, we need to overcome two difficulties: First, when trying to filter $Y$ on subsets of its coordinates, we do not know the true variance of $Y$, and thus cannot expect to obtain $\tilde{O}(\varepsilon)$ error. This is fixed with a bootstrapping method similar to that in [Kan18] to estimate the covariance of a Gaussian. In particular, we do not know $\mathbf{Var}[Y]$ a priori, but after we run the algorithm, we obtain an approximation to $v$, which gives an approximation to $\mathbf{Var}[Y]$. This in turn lets us get a better approximation to $v$ and a better approximation to $\mathbf{Var}[Y]$; and so on.

## 3 Experiments

For every experiment, we run 10 trials and plot the median value of the measurement. We shade the interquartile range around each measurement as a measure of the confidence of that measurement.

Each experiment was run on a computer with a 2.7 GHz Intel Core i5 processor with an 8GB 1867 MHz DDR3 RAM.

### 3.1 Robust Sparse Mean Estimation

The performance of robust estimation algorithms depend heavily on the noise model. The "hard" noise distributions for one algorithm may be easy for a different algorithm, if that one can identify and filter out the outliers. We therefore consider three different synthetic data distributions: two that demonstrate the $\varepsilon\sqrt{k}$ worst-case performance of other algorithms, and one that demonstrates the $\varepsilon\sqrt{\log(1/\varepsilon)}$ performance of our full algorithm.

The algorithms we consider are RME_sp, our algorithm; RME_sp_L, a version of our algorithm with only the linear filter and not the quadratic one; NP, the "naive pruning" algorithm that drops samples with obviously-outlier coordinates, then outputs the empirical mean; oracle, which is told exactly

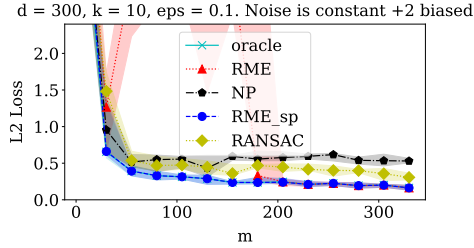

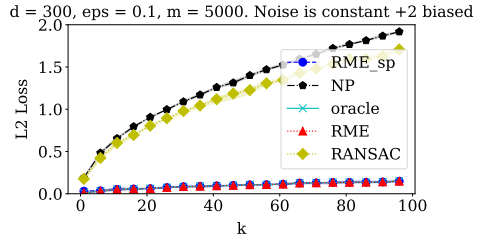

(a) Unlike `RANSAC`, our algorithm `RME_sp` can filter out the noise and match the oracle's performance. `RME` also matches the oracle, but needs more samples.

(b) For fixed $m$, as $k$ increases, `RANSAC` and `NP` both diverge from `RME` and `RME_sp`.

Figure 1: Constant-bias noise is easy for our algorithm, since it is caught by the linear filter.

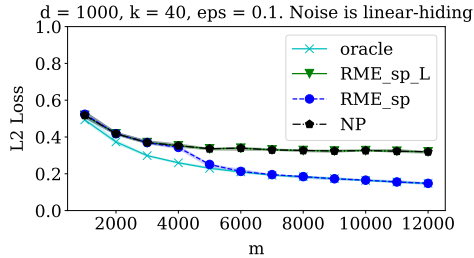

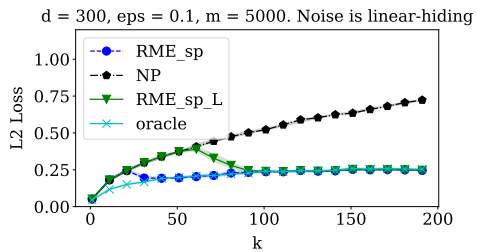

(a) With sufficiently many samples, the quadratic filter can filter out the noise, matching the oracle. The linear filter alone does not, even with a large number of samples.

(b) For $k \ll \sqrt{d}$, the linear filter alone does not filter out the noise, leading to an $\varepsilon\sqrt{k}$ dependence for `RME_sp_L`. Our algorithm `RME_sp` nearly matches `oracle`.

Figure 2: The linear-hiding noise model shows that the quadratic filter is necessary.

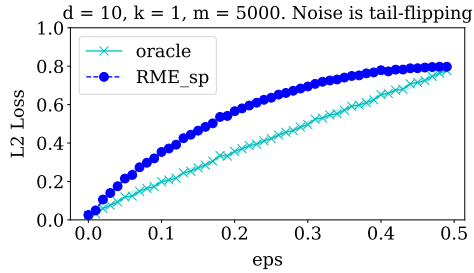

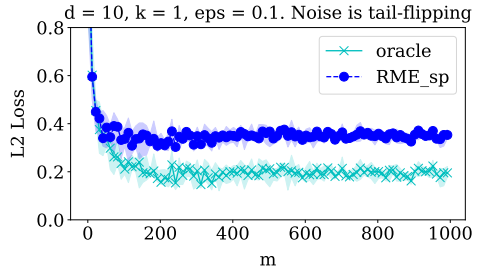

(a) This noise model gives $\Omega(\varepsilon\sqrt{\log(1/\varepsilon)})$ error to the oracle, and `RME_sp` is at most twice this.

(b) This gap persists regardless of $m$.

Figure 3: The flipping noise model demonstrates that the error can remain $\Omega(\varepsilon\sqrt{\log(1/\varepsilon)})$.

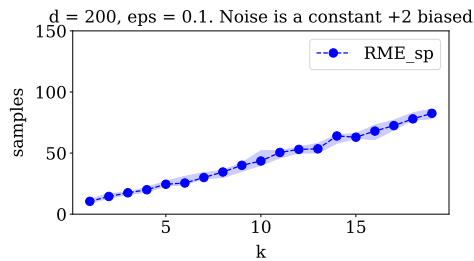

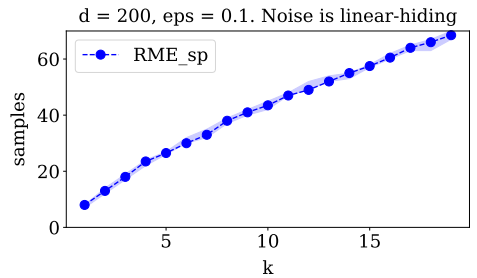

(a) The constant noise model is easy to remove and does not take many samples.

(b) The linear-hiding noise model is harder and requires more samples to get the same guarantee.

Figure 4: Sample complexity required to do well—in this case, 70% of errors being less than 1.2—depends on the noise model.

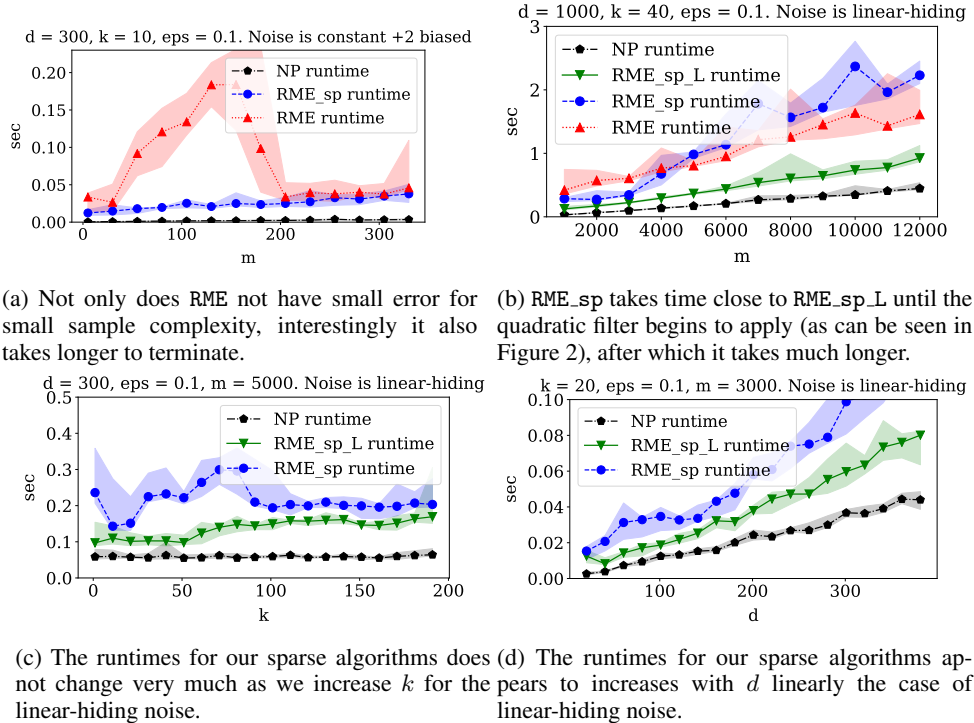

(a) Not only does `RME` not have small error for small sample complexity, interestingly it also takes longer to terminate.

(b) `RME_sp` takes time close to `RME_sp_L` until the quadratic filter begins to apply (as can be seen in Figure 2), after which it takes much longer.

(c) The runtimes for our sparse algorithms does not change very much as we increase $k$ for the linear-hiding noise.

(d) The runtimes for our sparse algorithms appears to increases with $d$ linearly the case of linear-hiding noise.

Figure 5: Runtimes for robust mean estimation.

which coordinates are inliers and outputs their empirical mean; `RME`, which applies the non-sparse robust mean estimation algorithm of [DKK+17]; and `RANSAC`, which computes the mean of a randomly chosen set of points, half the size of the entire set. One mean is preferred to another if it has more points in a ball of radius $\sqrt{d + \sqrt{d}}$ around it. For algorithms that have non-sparse outputs, we sparsify to the largest $k$ coordinates before measuring the $\ell_2$ distance to the true mean.

Our distributions are:

- **Constant-bias noise.** Noise that biases every coordinate consistently (e.g., if the outliers add 2 to every coordinate, or set every coordinate to $\mu_i + 1$) is difficult for naive algorithms (such as coordinate-wise median, `NP`, `RANSAC`) to deal with, but ideal for the linear filter. In Figure 1 we consider the noise that adds 2 to every coordinate.

- **Linear-hiding noise.** To demonstrate that the quadratic filter in our algorithm is necessary, we use the following data distribution. The inliers are drawn from $\mathcal{N}(0, I)$. The outliers are evenly split between two types: $\mathcal{N}(1_S, I)$ for some size-$k$ set $S$, and $\mathcal{N}(0, 2I - I_S)$. The diagonal of the empirical covariance does not reveal $S$, so our linear filter fails to prune anything, leading to $\varepsilon\sqrt{k}$ error for `RME_sp_L`; the quadratic filter successfully removes all the outliers. This is shown in Figure 2.

- **Flipping noise.** For both those types of noise, with sufficiently many samples our final algorithm will prune out essentially all the outliers; there also exist noise models where $\Omega(\varepsilon\sqrt{\log(1/\varepsilon)})$ noise will remain at all times. In Figure 3 we demonstrate this for the noise model that picks a $k$-sparse direction $v$, and replaces the $\varepsilon$ fraction of points furthest in the $-v$ direction with points in the $+v$ direction. In fact, for this noise even the `oracle` method also has $\Omega(\varepsilon\sqrt{\log(1/\varepsilon)})$ error from the missing points, but our algorithm has twice the error from the unfilterable added points.

**Discussion.** Matching our theoretical results, with sufficiently many samples the worst-case performance of `RME_sp` seems to be within a constant factor of the $O(\varepsilon\sqrt{\log(1/\varepsilon)})$ worst-case performance of `oracle`. This is not true for the naive algorithms `NP`, `RANSAC`, or the simplification `RME_sp_L` of our algorithm, which all have an $\varepsilon\sqrt{k}$ dependence. While our theoretical results show

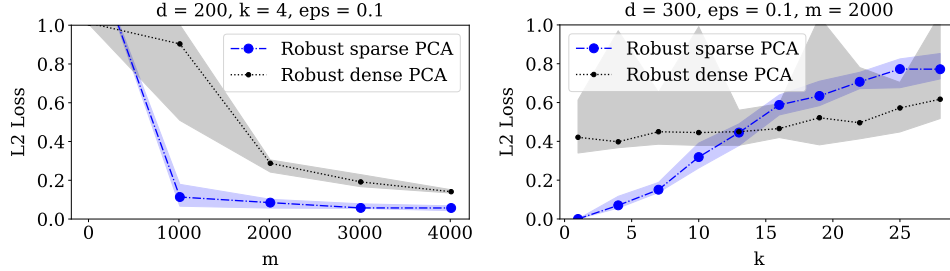

(a) The natural dense algorithm RDPCA requires more samples than the sparse algorithm to get error $< 0.1$

(b) For a fixed $m$, RSPCA performs better than RDPCA when $k < \sqrt{d}$ and then performs worse. until coming close to RDPCA. Note that the variance of RSPCA is smaller than that of RDPCA.

Figure 6: Sample complexity of RSPCA is better than RDPCA for smaller sparsity.

that $\widetilde{O}(k^2)$ samples suffice, the empirical results given in Figure 4 are consistent with $\widetilde{O}(k)$ being sufficient.

Our algorithm runs much faster than the ellipsoid based approach. For instance for $k = 10, d = 300, m = 50$ for the case of constant-biased noise our algorithm takes time $0.015$ seconds to finish. In comparison the very first iteration for the SDP-based solution takes $10$ seconds to solve with CVXOPT; the full ellipsoid-based algorithm, if implemented, would take many times that.

## 3.2 Robust Sparse PCA

In Figure 6 we compare our robust sparse PCA algorithm RSPCA to a dense algorithm RDPCA for robust PCA. RDPCA looks at the empirical covariance matrix and then in the direction of maximum variance robustly estimates standard deviation. The algorithm then filters points using a modified version of the linear filter from [DKK+17] and hence requires a sample complexity of $\widetilde{O}(d)$. For this algorithm, we only consider a single simple noise model. We draw outlier samples from $\mathcal{N}(0, I + uu^T)$ where $u$ has disjoint support from the true vector $v$.

The sparse algorithm seems to perform better than the dense algorithm for $k$ up to roughly $\sqrt{d}$; this is better than what we can prove, which is that it should be better up to at least $d^{1/4}$.

## 4 Conclusions

In this paper, we have presented iterative filtering algorithms for two natural and fundamental robust sparse estimation tasks: sparse mean estimation and sparse PCA. In both cases, our algorithms achieve near-optimal $\widetilde{O}(\varepsilon)$ error with sample complexity primarily dependent on the sparsity $k$, and only logarithmically on the ambient dimension $d$. Our theoretical guarantees are comparable to those of [BDLS17], with the significant advantage that our algorithms only use simple spectral techniques rather than the ellipsoid algorithm. This makes our algorithms practically viable and easy to implement. Our implementations perform essentially as expected: in sparse settings they require significantly fewer samples than dense robust estimation, and have accuracy avoiding the $\sqrt{k}$ dependence of common benchmark techniques like RANSAC.

## 5 Acknowledgements

The authors would like to thank the following sources of support.

Ilias Diakonikolas was supported by the NSF Award CCF-1652862 (CAREER) and a Sloan Research Fellowship. Sushrut Karmalkar was supported by NSF Award CNS-1414023. Daniel Kane was supported by NSF Award CCF-1553288 (CAREER) and a Sloan Research Fellowship. Eric Price was supported in part by NSF Award CCF-1751040 (CAREER). A part of this work was performed when Alistair Stewart was a postdoctoral researcher at USC.

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
