[Supplementary Material]

**Supplementary Material**

 # A   Robust Sparse Mean Estimation

373 In this section, we prove correctness of Algorithm 1 establishing Theorem 1.2. For completeness,
374 we restate a formal version of this theorem:

375 **Theorem A.1.** *Let $D \sim \mathcal{N}(\mu, I)$ be an identity covariance Gaussian distribution on $\mathbb{R}^d$ with un-*
376 *known $k$-sparse mean vector $\mu$, and $\varepsilon, \tau > 0$. Let $S$ be an $\varepsilon$-corrupted set of samples from $D$ of size*
377 $N = \widetilde{\Omega}(k^2 \log(d/\tau)/\varepsilon^2)$. *There exists an efficient algorithm that, on input $S$, $k$, $\varepsilon$, and $\tau$, returns a*
378 *mean vector $\widehat{\mu}$ such that with probability at least $1 - \tau$ it holds $\|\widehat{\mu} - \mu\|_2 = O(\varepsilon\sqrt{\log(1/\varepsilon)})$.*

379 ## A.1   Preliminaries

380 We will use the following notation and definitions.

381 **Basic Notation**   For $n \in \mathbb{Z}_+$, let $[n] \overset{\text{def}}{=} \{1, 2, \ldots, n\}$. Throughout this paper, for $v =$
382 $(v_1, \ldots, v_d) \in \mathbb{R}^d$, we will use $\|v\|_2$ to denote its Euclidean norm. If $M \in \mathbb{R}^{d \times d}$, we will use
383 $\|M\|_2$ to denote its spectral norm, $\|M\|_F$ to denote its Frobenius norm, and $\text{tr}[M]$ to denote its
384 trace. We will also let $\preceq$ and $\succeq$ denote the PSD ordering on matrices. For a finite multiset $S$, we
385 will write $X \in_u S$ to denote that $X$ is drawn from the empirical distribution defined by $S$. Given
386 finite multisets $S$ and $S'$ we let $\Delta(S, S')$ be the size of the symmetric difference of $S$ and $S'$ divided
387 by the cardinality of $S$.

388 For $v \in \mathbb{R}^d$ and $S \subseteq [d]$, let $v_S$ be the vector with $(v_S)_i = v_i$, $i \in S$, and $(v_S)_i = 0$ otherwise.
389 We denote by $h_k(v)$ the thresholding operator that keeps the $k$ entries of $v$ with largest magnitude
390 (breaking ties arbitrarily) and sets the rest to 0. For $M \in \mathbb{R}^{d \times d}$ and $U \subseteq [d]$, let $M_U$ denote the
391 matrix $M$ restricted to the $U \times U$ sub-matrix. For $W \subseteq [d] \times [d]$, then we will use $M_{(W)}$ to denote
392 the matrix $M$ restricted to the elements whose entries are in $W$.

393 Let $\delta_{ij}$ denote the Kronecker delta function. We will denote $\text{erfc}(z) = (2/\sqrt{\pi}) \int_z^\infty e^{-t^2} dt$. The
394 notation $\widetilde{O}(\cdot)$ and $\widetilde{\Omega}(\cdot)$ hides logarithmic factors in the argument.

395 ## A.2   Proof of Theorem A.1

396 In this section, we describe and analyze our algorithm establishing Theorem A.1. We start by for-
397 malizing the set of deterministic conditions on the good data under which our algorithm succeeds:

398 **Definition A.2.** *Fix $0 < \varepsilon, \tau < 1$ and $k \in \mathbb{Z}_+$. A multiset $G$ of points in $\mathbb{R}^d$ is $(\varepsilon, k, \tau)$-good with*
399 *respect to $\mathcal{N}(\mu, I)$ if, for $X \in_u G$ and $Y \sim \mathcal{N}(\mu, I)$, the following conditions hold:*

400    *(i)  For all $i \in [d]$, $|\mathbf{E}[X_i] - \mu_i| \leq \varepsilon/k$, and for all $i, j \in [d]$, $|\mathbf{E}[(X_i - \mu_i)(X_j - \mu_j)] - \delta_{ij}| \leq$*
401       $\varepsilon/k$.

402    *(ii) For all $x \in G$ and $i \in [d]$, we have $|x_i - \mu_i| \leq O(\sqrt{\log(d|G|/\tau)})$.*

403    *(iii) For all $2k^2$-sparse unit vectors $v \in \mathbb{R}^d$, we have that:*

404       *(a)  $|\mathbf{E}[v \cdot (X - \mu)]| \leq O(\varepsilon)$,*
405       *(b)  $|\mathbf{E}[(v \cdot (X - \mu))^2] - 1| \leq O(\varepsilon)$, and*
406       *(c)  For all $T \geq 6$, $\mathbf{Pr}[|v \cdot (X - \mu)| \geq T] \leq 3 \cdot \text{erfc}(T/\sqrt{2}) + \varepsilon^2/\left(T^2 \ln\left(k \ln(d|G|/\tau)\right)\right)$.*

407    *(iv) For all homogeneous\* degree-2 polynomials $p$ with $\mathbf{Var}[p(Y)] = 1$ and at most $k^2$ terms,*
408       *we have that:*

409       *(a)  $|\mathbf{E}[p(X)] - \mathbf{E}[p(Y)]| \leq O(\varepsilon\sqrt{\mathbf{Var}[p(Y)]}) = O(\varepsilon)$, and,*
410       *(b)  For all $T \geq 5$, $\mathbf{Pr}\left[|p(X) - \mathbf{E}[p(Y)]| \geq T\right] \leq 3 \exp(-T/4) + \varepsilon^2/(T \ln^2 T)$.*

Our first lemma says that a sufficiently large set of samples from $\mathcal{N}(\mu, I)$ is good with high probability:

**Lemma A.3.** *A set of $N = \widetilde{\Omega}\left(k^2 \log(d/\tau)/\varepsilon^2\right)$ samples from $\mathcal{N}(\mu, I)$ is $(\varepsilon, k, \tau)$-good (with respect to $\mathcal{N}(\mu, I)$) with probability at least $1 - \tau$.*

*Proof.* Let $G$ be a set of $N = \widetilde{\Omega}\left(k^2 \log(d/\tau)/\varepsilon^2\right)$ i.i.d. samples drawn from $\mathcal{N}(\mu, I)$. We will show that each of Conditions (i)-(iv) hold with probability at least $1 - \tau/5$. The lemma then follows by a union bound.

**Proof of (i):** To establish (i), let $\mu^G := \mathbf{E}_{X \in_u G}[X]$ and note that the random variable $N\mu^G$ is distributed as $\mathcal{N}(N \cdot \mu, N \cdot I)$. Hence, $\mu^G$ has independent coordinates with $N\mu_i^G \sim \mathcal{N}(N \cdot \mu_i, N)$. By standard Gaussian tail bounds, we have that $\mathbf{Pr}\left[\left|N(\mu_i^G - \mu_i)\right| \geq T\sqrt{N}\right] \leq 2 \cdot \exp(-T^2/2)$. Setting $T/\sqrt{N} = \varepsilon/k$ gives that $\mathbf{Pr}\left[|\mu_i^G - \mu_i| \geq \varepsilon/k\right] \leq 2 \cdot \exp(-N\varepsilon^2/(2k^2)) \leq \tau/(10d)$. By a union bound over all $i \in [d]$, it follows that

$$\mathbf{Pr}\left[\exists i \in [d] : |\mu_i^G - \mu_i| \geq \varepsilon/k\right] \leq \tau/10 .$$

This completes the proof of the first part of (i).

For the second part of (i), we will show that with probability at least $1 - \tau/10$ we have that for all $i, j \in [d]$, $\left|\mathbf{E}\left[(X_i - \mu_i)(X_j - \mu_j)\right] - \delta_{ij}\right| \leq \varepsilon/k$. We will need the following simple technical fact:

**Fact A.4** (see, e.g., [LM00]). *Let $Y_i$ be iid standard univariate Gaussians and $a_i \geq 0$, $i \in [m]$. If $Z = \sum_{i=1}^{m} a_i(Y_i^2 - 1)$, then for any $x \geq 0$ the following hold:*

$$\mathbf{Pr}\left[Z \geq 2\|a\|_2\sqrt{x} + 2\|a\|_\infty x\right] \leq \exp(-x) , \tag{1}$$

*and*

$$\mathbf{Pr}\left[Z \leq -2\|a\|_2\sqrt{x}\right] \leq \exp(-x) . \tag{2}$$

We start with the case that $i = j$. Note that the random variable $N \cdot \mathbf{E}_{X \in_u G}\left[(X_i - \mu_i)^2\right]$ follows a $\chi^2$-distribution with $N$ degrees of freedom, i.e., it is the sum of $N$ independent squared standard Gaussians. An application of Equation (1) implies that for all $x \geq 0$ we have:

$$\mathbf{Pr}\left[\left|N \cdot \mathbf{E}_{X \in_u G}\left[(X_i - \mu_i)^2\right] - N\right| \geq 2\sqrt{Nx} + 2x\right] \leq \exp(-x).$$

Setting $x := N\varepsilon^2/(9k^2)$, we get that

$$\mathbf{Pr}\left[\left|\mathbf{E}_{X \in_u G}\left[(X_i - \mu_i)^2\right] - 1\right| \geq 2\varepsilon/(3k) + 2\varepsilon^2/(9k^2)\right] \leq \exp\left(-N\varepsilon^2/(9k^2)\right) \leq \tau/(10d^2).$$

We now analyze the case that $i \neq j$. Let $Y \sim \mathcal{N}(\mu, I)$. Note that for $i \neq j$, $i, j \in [d]$, we have that

$$(Y_i - \mu_i)(Y_j - \mu_j) = \left(\frac{(Y_i - \mu_i)}{2} + \frac{(Y_j - \mu_j)}{2}\right)^2 - \left(\frac{(Y_i - \mu_i)}{2} - \frac{(Y_j - \mu_j)}{2}\right)^2 .$$

Since $\frac{(Y_i - \mu_i)}{2} + \frac{(Y_j - \mu_j)}{2}$ and $\frac{(Y_i - \mu_i)}{2} - \frac{(Y_j - \mu_j)}{2}$ are independent and distributed as $\mathcal{N}(0, 1/2)$, for $i \neq j$, the random variable $N \cdot \mathbf{E}_{X \in_u G}\left[(X_i - \mu_i)(X_j - \mu_j)\right]$ is distributed as the difference of a sum of $N$ independent squared zero-mean Gaussians with variance $1/2$, and another such sum. This random variable has expectation $0$ and once again, by Equation (1) applied with $a_i = 1/2$, it follows that

$$\mathbf{Pr}\left[\left|N \cdot \mathbf{E}_{X \in_u G}\left[(X_i - \mu_i)(X_j - \mu_j)\right]\right| \geq 2\sqrt{Nx} + x\right] \leq \exp(-x) .$$

Setting $x := N\varepsilon^2/(9k^2)$ as above gives that

$$\mathbf{Pr}\left[\left|\mathbf{E}_{X \in_u G}\left[(X_i - \mu_i)(X_j - \mu_j)\right]\right| \geq \varepsilon/k\right] \leq \tau/(10d^2).$$

A union bound over all $i, j \in [d]$ implies that

$$\mathbf{Pr}\left[\exists i, j \in [d] : \left|\mathbf{E}_{X \in_u G}\left[(X_i - \mu_i)(X_j - \mu_j)\right] - \delta_{ij}\right| \geq \varepsilon/k\right] \leq \tau/10.$$

This gives the second part of (i). By a union bound, Condition (i) holds with probability at least $1 - \tau/5$.

**Proof of (ii):** For $Y \sim \mathcal{N}(\mu, I)$, the standard Gaussian tail bound gives $\mathbf{Pr}\left[|Y_i - \mu_i| \geq T\right] \leq 2\exp\left(-T^2/2\right)$. Setting $T = \sqrt{2\ln(10Nd/\tau)}$ implies that $\mathbf{Pr}\left[|Y_i - \mu_i| \geq T\right] \leq \tau/(10Nd)$. By a union bound, the desired upper bound holds for all $i \in [d]$ and all $N$ samples with probability at least $1 - \tau/10$.

**Proof of (iii):** To establish (iii), we first prove that Conditions (iii)(a)-(c) hold for any fixed unit vector $v$ and threshold $T$ with sufficiently high probability, and then take a union bound over a net of $2k^2$-sparse unit vectors and thresholds.

To avoid clutter in the notation, we will denote $\delta \overset{\text{def}}{=} \frac{\varepsilon^2}{\ln(k\ln(Nd/\tau))}$, so that the second term in the RHS of Condition (iii)(c) is equal to $\delta/T^2$.

We start by proving the following claim:

**Claim A.5.** *For any unit vector $v$ in $\mathbb{R}^d$ and threshold $T \geq \Omega(1)$ with probability at least $1 - \exp\left(-\Omega\left(\frac{N\delta}{\log(1/\delta)}\right)\right)$, we have that (a) $|\mathbf{E}_{X \in_u G}[v \cdot (X - \mu)]| \leq O(\varepsilon)$, (b) $|\mathbf{E}_{X \in_u G}[(v \cdot (X - \mu))^2] - 1| \leq O(\varepsilon)$, and (c) $\mathbf{Pr}_{X \in_u G}[|v \cdot (X - \mu)| \geq T] \leq (5/2) \cdot \mathrm{erfc}(T/\sqrt{2}) + \delta/(2T^2)$.*

*Proof.* To prove (a), note that for each fixed unit vector $v \in \mathbb{R}^d$, $N\mathbf{E}_{X \in_u G}[v \cdot (X - \mu)]$ is distributed as $\mathcal{N}(0, N)$. By standard Gaussian tail bounds, we have that

$$\mathbf{Pr}\left[|\mathbf{E}_{X \in_u G}[v \cdot (X - \mu)]| \geq \varepsilon\right] \leq 2 \cdot \exp(-N\varepsilon^2/2) \ll \exp\left(-\Omega\left(N\delta/\log(1/\delta)\right)\right) ,$$

where the last inequality follows from the fact that $\delta \ll \varepsilon^2$.

To prove (b), note that for each fixed unit vector $v \in \mathbb{R}^d$ the random variable $N \cdot \mathbf{E}_{X \in_u G}[(v \cdot (X - \mu))^2]$ follows a $\chi^2$-distribution with parameter $N$. By Equation (1), we get

$$\mathbf{Pr}\left[|N \cdot \mathbf{E}_{X \in_u G}[(v \cdot (X - \mu))^2] - N| \geq 2\sqrt{Nx} + 2x\right] \leq \exp(-x) ,$$

for $x \geq 0$. Applying the above inequality for $x := N\varepsilon^2/9$, we get

$$\mathbf{Pr}\left[|\mathbf{E}_{X \in_u G}[(v \cdot (X - \mu))^2] - 1| \geq 2\varepsilon/3 + (2/9)\varepsilon^2\right] \leq \exp\left(-N\varepsilon^2/9\right) .$$

To prove (c), we start by noting that, for any fixed unit vector $v$ and $Y \sim \mathcal{N}(\mu, I)$, $v \cdot (Y - \mu)$ is a standard univariate Gaussian, and therefore $\mathbf{Pr}[|v \cdot (Y - \mu)| \geq T] = 2\mathrm{erfc}(T/\sqrt{2})$. Let

$$Q(T) \overset{\text{def}}{=} (5/2)\mathrm{erfc}(T/\sqrt{2}) + \delta/(2T^2) .$$

Observe that $N \cdot \mathbf{Pr}_{X \in_u G}[|v \cdot (X - \mu)| \geq T]$ is a sum of $N$ independent Bernoulli random variables each with mean $2\mathrm{erfc}(T/\sqrt{2})$. An application of the Chernoff bound and the fact that $Q(T) \geq (5/4)\left[2\mathrm{erfc}(T/\sqrt{2})\right]$ gives that $\mathbf{Pr}_{X \in_u G}[|v \cdot (X - \mu)| \geq T] \geq Q(T)$ holds with probability at most $\exp\left(-\frac{NQ(T)}{60}\right)$.

We choose $T'$ to satisfy $\mathrm{erfc}(T') = \delta^2/(4T'^4)$, which implies that $T' = \Theta(\sqrt{\ln(1/\delta)})$. We break the analysis into two cases: $T \leq T'$ or $T > T'$.

If $T \leq T'$, then $Q(T) \geq Q(T') \geq \delta/(2T'^2) = \Omega(\frac{\delta}{\log(1/\delta)})$ and the above upper bound of $\exp\left(-\frac{NQ(T)}{60}\right)$ on the desired probability gives (c).

If $T > T'$, we have that $\mathrm{erfc}(T/\sqrt{2}) \leq \delta^2/(4T^4)$. In this case, we require a more precise version of the Chernoff bound, which bounds from above the probability of the event $\mathbf{Pr}_{X \in_u G}[|v \cdot (X - \mu)| \geq T] \geq Q(T)$ by $\exp\left(-N \cdot D_{KL}(Q(T)||2\mathrm{erfc}(T/\sqrt{2}))\right)$, where $D_{KL}(p||q)$ denotes the KL-divergence between the Bernoulli random variables with probabilities $p$ and $q$.

Let $p = \delta/(2T^2)$, $q = 2\mathrm{erfc}(T/\sqrt{2})$, and note that $q \le p^2$ or $p/q \ge q^{-1/2}$. We can now bound from below the KL-divergence by $\delta/10$, as follows:

$$
\begin{aligned}
D_{KL}(Q(T)\|q) &\ge D_{KL}(p\|q) = p \ln(p/q) + (1-p) \ln \left((1-p)/(1-q)\right) \\
&\ge p \ln(p/q) - \ln(1-p) \ge p \left(\ln(p/q) - 1 - O(p)\right) \\
&\ge \delta/(2T^2) \cdot \left(\ln(1/q^{1/2}) - 1 - O(p)\right) \\
&\ge \delta/(2T^2) \cdot \left(T^2/8 - O(1)\right) \ge \delta/10 \,,
\end{aligned}
$$

where we used the assumption that $T$ is at least a sufficiently large universal constant. Thus, we have that $\mathbf{Pr}_{X \in_u G}[|v \cdot (X - \mu)| \ge T] \ge Q(T)$ with probability at most $\exp(-\Omega(N\delta))$ in this case. This completes the proof of (c).

By a union bound, all events hold with probability at least $1 - \exp\left(-\Omega\left(\frac{N\delta}{\log(1/\delta)}\right)\right)$, completing the proof of Claim A.5. $\qquad\square$

We now define a cover over all $2k^2$-sparse vectors as well as the possible values of $T$, and take a union bound over the product. To this end, let

$$
R \stackrel{\text{def}}{=} \Theta\left(k \cdot \sqrt{\log(Nd/\tau)}\right)
$$

be such that by (ii) we have $\|x - \mu\|_\infty \le \frac{R}{\sqrt{2}k}$, for $x \in G$.

For each set $U \subseteq [d]$ of coordinates of size $2k^2$, let $\mathcal{C}_U$ be an $\varepsilon/R^2$-cover, in $\ell_2$-norm, of the set of unit vectors supported on $U$ (i.e., with all non-zero coordinates in $U$). Such a cover exists with $|\mathcal{C}_U| \le O\left(R^2/\varepsilon\right)^{2k^2}$. Let $\mathcal{C}$ be the union of $\mathcal{C}_U$ over all sets $U$ of coordinates of size $2k^2$. Then we have that

$$
|\mathcal{C}| \le \binom{d}{2k^2} \cdot O(R^2/\varepsilon)^{2k^2} \le O\left(dR^2/\varepsilon\right)^{2k^2} \,.
$$

Let $\mathcal{T} := \{\sqrt{i\varepsilon} \mid i \in \mathbb{Z}_+, 0 \le i \le R^2/\varepsilon^2\}$ be a net over thresholds $T$. Note that $|\mathcal{C}| \cdot |\mathcal{T}| \le O(dR^2/\varepsilon)^{2k^2+2}$. By a union bound, Claim A.5 holds for all $v \in \mathcal{C}$ and $T \in \mathcal{T}$ except with probability at most

$$
\begin{aligned}
&O\left((dk^2/\varepsilon) \log(Nd/\tau)\right)^{2k^2+2} \cdot \exp(\Omega(-N\delta/\log(1/\delta)) \\
&= \exp\left(O(k^2 \log\left(dk \log(d/\tau)/\varepsilon\right)) - \Omega\left(N\varepsilon^2/\log^3(k/\varepsilon \log(d/\tau))\right)\right) \le \tau/10 \,,
\end{aligned}
$$

where we used the fact that $N = \widetilde{\Omega}\left(k^2 \log(d/\tau)/\varepsilon^2\right)$. It remains to prove (iii) assuming this event holds.

By definition, for any $k^2$-sparse unit vector $v \in \mathbb{R}^d$, there exists a $v' \in \mathcal{C}$ such that $\|v' - v\|_2 \le \varepsilon/R^2$ and such that $v' - v$ is also $k^2$-sparse. Thus, for any $x \in G$, we have

$$
\begin{aligned}
|v \cdot (x - \mu) - v' \cdot (x - \mu)| &\le \|v' - v\|_1 \|x - \mu\|_\infty \\
&\le \sqrt{2}k \|v' - v\|_2 R/\sqrt{2}k \le \varepsilon/R \,.
\end{aligned}
$$

Therefore, for the mean we have that $|\mathbf{E}_{X \in_u G}[v \cdot X]| \le |\mathbf{E}_{X \in_u G}[v' \cdot X]| + \frac{\varepsilon}{R} \le O(\varepsilon)$. This gives Condition (iii)(a).

To establish Condition (iii)(b), we note that for any $x \in G$, we have

$$
\begin{aligned}
|(v \cdot (x - \mu))^2 - (v' \cdot (x - \mu))^2| &\le O\left(|v \cdot (x - \mu) - v' \cdot (x - \mu)| \left(|v \cdot (x - \mu)| + |v' \cdot (x - \mu)|\right)\right) \\
&\le O(\varepsilon/R) \cdot O(k \cdot R/k) \\
&\le O(\varepsilon) \,,
\end{aligned}
$$

where the second line uses the fact that $|v \cdot (x - \mu)| \le \|v\|_1 \|x - \mu\|_\infty \le k \|x - \mu\|_\infty \le R$. Therefore, we have that $|\mathbf{E}_{X \in_u G}[(v \cdot (X - \mu))^2] - 1| \le |\mathbf{E}_{X \in_u G}[(v' \cdot (X - \mu))^2] - 1| + O(\varepsilon) = O(\varepsilon)$. This gives Condition (iii)(b).

We now prove Condition (iii)(c). Consider the event $\{x \in G : |v \cdot (x - \mu)| \geq T\}$ for $T \geq \sqrt{2 \ln(1/\varepsilon) + 2}$. First note that this event is contained in the event $\{x \in G : |v' \cdot (x - \mu)| \geq T - \varepsilon/R\}$. Moreover, note that the event is empty, unless $T \leq \|v\|_1 \|x - \mu\|_\infty \leq R$, in which case $(T - \varepsilon/R)^2 \geq T^2 - 2\varepsilon$. Therefore, by the definition of $\mathcal{T}$, there is a $T' \in \mathcal{T}$ with $T^2 - 2\varepsilon \leq T'^2 \leq (T - \varepsilon/R)^2$. Then we have

$$
\begin{aligned}
\mathbf{Pr}_{X \in_u G}[|v \cdot (X - \mu)| \geq T] &\leq \mathbf{Pr}_{X \in_u G}[|v' \cdot (X - \mu)| \geq T - \varepsilon/R] \\
&\leq \mathbf{Pr}_{X \in_u G}[|v' \cdot (X - \mu)| \geq T'] \\
&\leq 5\mathrm{erfc}(T')/2 + \delta/(2T'^2) \\
&\leq 5\mathrm{erfc}\left(\sqrt{T^2 - 2\varepsilon}\right)/2 + \delta/(2(T^2 - 2\varepsilon)) \\
&= (5/(2\sqrt{2\pi})) \int_{\sqrt{T^2 - 2\varepsilon}}^{\infty} \exp(-x^2/2)dx + \delta/T^2 \\
&= (5/(2\sqrt{2\pi})) \int_{T}^{\infty} \exp(-(y^2 - 2\varepsilon)/2)(y/\sqrt{y^2 - 2\varepsilon})dy + \delta/T^2 \\
&= (5/(2\sqrt{2\pi})) \int_{T}^{\infty} \exp(\varepsilon) \exp(-y^2/2)(1 + O(\varepsilon))dy + \delta/T^2 \\
&\leq (5/(2\sqrt{2\pi})) \int_{T}^{\infty} (1 + O(\varepsilon)) \exp(-y^2/2)dy + \delta/T^2 \\
&\leq 3\mathrm{erfc}(T/\sqrt{2}) + \delta/T^2 ,
\end{aligned}
$$

where the third line follows from Claim A.5(c) applied for $(v', T')$. This completes the proof of Condition (iii)(c).

**Proof of (iv):** At a high-level, the proof is similar to that of Condition (iii) above. We start by proving that Conditions (iv)(a)-(b) hold for any fixed degree-2 polynomial and threshold $T$ with sufficiently high probability, and then take a union bound over a net of $k^2$-sparse $p(x)$ and $T$.

Note that a homogeneous degree-2 polynomial can be written as $p(x) = (x - \mu)^T A(x - \mu)$, for a symmetric matrix $A$, in which case we have $\mathbf{E}_{Y \sim \mathcal{N}(\mu, I)}[p(Y)] = \mathrm{Tr}(A)$ and $\mathbf{Var}_{Y \sim \mathcal{N}(\mu, I)}[p(Y)] = \|A\|_F^2$.

We start by establishing the following claim:

**Claim A.6.** *Let $Y \sim \mathcal{N}(\mu, I)$. Given a homogeneous degree-2 polynomial $p(x)$ with $\mathbf{Var}[p(Y)] = 1$ and $T$ with $4 \leq T \leq R \stackrel{\text{def}}{=} \Theta(k \cdot \sqrt{\log(Nd/\tau)})$, we have that: (a) $|\mathbf{E}_{X \in_u G}[p(X)] - \mathbf{E}[p(Y)]| \leq O(\varepsilon)$, and (b) $\mathbf{Pr}_{X \in_u G}[|p(X) - \mathbf{E}[p(Y)]| \geq T] \leq 2\exp(-T/4) + \varepsilon^2/(2T \ln^2 T)$, except with probability at most $\exp(-\Omega(N\varepsilon^2/\ln^2(R/\varepsilon)))$.*

*Proof.* By diagonalizing $A$, we can write $p(Y) = c + \sum_{i=1}^{d} a_i Z_i^2$, where the $Z_i$ are independent and distributed as $\mathcal{N}(0, 1)$ and $c, a_i$ are real coefficients with $\sum_i a_i^2 = \|A\|_F^2 = 2$. Note that $N\mathbf{E}[p(X)]$ is a sum of $Nd$ independent squared Gaussians, each of which has variance at most $\|A\|_F^2 = 1$ and the $\ell_2$-norm of all their variances is $\sqrt{N}\|A\|_F = \sqrt{N}$. Equation (1) gives that $\mathbf{Pr}[|N\mathbf{E}[p(X)] - N\mathbf{E}[p(Y)]| \geq 2\sqrt{Nx} + 2x] \leq \exp(-x)$, for $x \geq 0$. Taking $x := N\varepsilon^2$, we obtain that

$$
\mathbf{Pr}\left[|\mathbf{E}[p(X)] - \mathbf{E}[p(Y)]| \geq 2\varepsilon + 2\varepsilon^2\right] \leq \exp(-N\varepsilon^2).
$$

This shows (a).

We proceed to prove (b). By Equation (1) applied for a single sample, we have that $\mathbf{Pr}[|p(Y) - \mathbf{E}[p(Y)]| \geq 2\sqrt{x} + 2x] \leq \exp(-x)$ for $x \geq 0$. Taking $x := T$ for $T \geq 4$, we have $2\sqrt{T} \leq T$, and so

$$
\mathbf{Pr}[|p(Y) - \mathbf{E}[p(Y)]| \geq T] \leq \exp(-T/4).
$$

Note that $N\mathbf{Pr}[|p(X) - \mathbf{E}[p(Y)]| \geq T]$ is a sum of $N$ independent Bernoulli random variables each with expectation at most $\exp(-T/4)$. Let

$$
Q(T) \stackrel{\text{def}}{=} 2\exp(-T/4) + \varepsilon^2/(2T \ln^2 T) .
$$

533  Since $Q(T) \geq 2\mathbf{Pr}[|p(Y) - \mathbf{E}[p(Y)]| \geq T]$, by the multiplicative Chernoff bound we have that
534  $\mathbf{Pr}[|p(X) - \mathbf{E}[p(Y)]| \geq T] \leq Q(T)$, except with probability at most $\exp(-NQ(T)/6)$.

535  Let $T'$ be such that $\exp(-T'/6) = \varepsilon^2/(2T' \ln^2(T'))$. Note that $T' = \Theta(\log(1/\varepsilon))$.
536  For $T \leq T'$, we have that $Q(T) \geq \varepsilon^2/(T' \ln^2 T')$, and so $\exp(-NQ(T)/6) \geq$
537  $\exp(\Omega(-N\varepsilon/\log(1/\varepsilon)(\log\log(1/\varepsilon))^2))$.

538  For $T \geq T'$, note that $\varepsilon^2/(2T \ln^2(T)) \geq \exp(-T/6)$. Again we need to use a more explicit version
539  of the Chernoff bound, which gives that $\mathbf{Pr}_{X \in_u G}[|v \cdot (X - \mu)| \geq T] \geq Q(T)$ with probability at
540  most $\exp(-ND_{KL}(Q(T)||\exp(-T/4)))$.

541  When $T' \leq T \leq R$, $p = \varepsilon^2/(2T \ln^2 T)$, and $q = 2\exp(-T/4)$, we obtain

$$
\begin{aligned}
D_{KL}(Q(T)||q) &\geq D_{KL}(p||q) = p\ln(p/q) + (1-p)\ln((1-p)/(1-q)) \\
&\geq p\ln(p/q) - \ln(1-p) \\
&\geq p(\ln(p/q) - 1 - O(p)) \\
&= (\varepsilon^2/(2T \ln^2 T))(\ln(p/\exp(-T/4)) - 1 - O(p)) \\
&\geq (\varepsilon^2/(2T \ln^2 T))(\ln(\exp(T/6)) - 1 - O(p)) \\
&\geq (\varepsilon^2/(2T \ln^2 T)) \cdot (T/7) \\
&\geq \varepsilon^2/(14 \ln^2 T) \geq \varepsilon^2/(14 \ln^2 R) \;,
\end{aligned}
$$

542  where we used the fact that $\Omega(1) \leq T \leq R$. Thus, it follows that $\mathbf{Pr}_{X \in_u G}[|v \cdot (X - \mu)| \geq T] \geq$
543  $Q(T)$ with probability at most $\exp(-\Omega(N\varepsilon^2/\ln^2 R))$ in this case. In either case, by a union bound,
544  the claim holds except with probability $\exp(-\Omega(N\varepsilon^2/\ln^2(R/\varepsilon)))$. This completes the proof of (b)
545  and of Claim A.6.  □

546  It remains to construct a cover of $k^2$-sparse homogeneous degree-2 polynomials which have at most
547  $k^2$ terms and $\mathbf{Var}[p(Y)] = 1$. Let $U$ be the set of $k^2$ monomials $x_i x_j$, for $1 \leq i, j \leq d$. We construct
548  a cover $\mathcal{C}_U$ of polynomials with terms only in the monomials in $U$ as follows: We take a cover of
549  unit vectors in $\mathbb{R}^{k^2}$ to within $\ell_2$-norm $\varepsilon/R^2$ and use the coordinates of each vector as the coefficients
550  of the corresponding monomial. Thus, we can take $|\mathcal{C}_U| = 2^{O(k^2)}$. Then we let $\mathcal{C}$ be the union of
551  $\mathcal{C}_U$ for all sets of $k^2$ monomials $U$. We therefore have that $|\mathcal{C}| \leq \binom{d}{k^2} \cdot O(R^2/\varepsilon)^{k^2} \leq O(dR^2/\varepsilon)^{k^2}$.

552  Let $\mathcal{T} = \{i\varepsilon : i \in \mathbb{Z}_+, 0 \leq i \leq R^2/\varepsilon^2\}$. Thus, $|\mathcal{C}| \cdot |\mathcal{T}| \leq O(dR^2/\varepsilon)^{k^2+1}$. By a union bound,
553  Claim A.6 holds for all $p \in \mathcal{C}$ and $T \in \mathcal{T}$, except with probability at most

$$
\begin{aligned}
&O(dk^2 \log(Nd/\tau)/\varepsilon)^{k^2+1} \cdot \exp(-\Omega(N\varepsilon^2/\ln^2(R/\varepsilon))) \\
&= \exp\left(O(k^2 \log(dk \log(N/\tau)/\varepsilon)) - \Omega(N\varepsilon^2/\ln^2(k \log(Nd/\tau)/\varepsilon))\right) \leq \tau/10 \;,
\end{aligned}
$$

554  where we used the fact that $N = \widetilde{\Omega}\left(k^2 \log(d/\tau)/\varepsilon^2\right)$. It remains to prove (iv) assuming this event
555  holds.

556  Consider any homogeneous degree-2 polynomial $p(x)$ with at most $k^2$ terms and $\mathbf{Var}[p(Y)] =$
557  1. By construction of the cover, there is a polynomial $p'(x) \in \mathcal{C}$ such that the total number of
558  monomials appearing in either $p(x)$ or $p'(x)$ is at most $k^2$, and if we write $p(x) = (x-\mu)^T A(x-\mu)$
559  and $p'(x) = (x-\mu)^T A'(x-\mu)$ for symmetric matrices $A, A'$, then $\|A - A'\|_F \leq \varepsilon/R^2$. Let $U'$ be
560  the set of coordinates appearing in either $p(x)$ and $p'(x)$ and note that $|U'| \leq 2k^2$. For $x \in G$, we
561  have

$$
\begin{aligned}
|p(x) - p'(x)| &= |(x-\mu)^T (A - A')(x-\mu)| \\
&= |(x-\mu)_{U'}^T (A - A')(x-\mu)_{U'}| \\
&\leq \|(x-\mu)_{U'}\|_\infty^2 \|A - A'\|_2 \\
&\leq R^2 \cdot \varepsilon/R^2 \leq \varepsilon \;.
\end{aligned}
$$

562  Therefore, we have that $|\mathbf{E}[p(X)] - \mathbf{E}[p(Y)]| \leq |\mathbf{E}[p'(X)] - \mathbf{E}[p'(Y)]| + 2\varepsilon \leq O(\varepsilon)$, since Claim A.6
563  holds for $p'(x)$. We have thus established Condition (iv)(a).

564 To show Condition (iv)(b), consider the event $\{x \in G : |p(x)| \geq T\}$ for $T > 5$. Let $T' \in \mathcal{T}$ be such
565 that $T - 2\varepsilon \leq T' \leq T - \varepsilon$. Then, $|p(x)| \geq T$ implies that $|p'(x)| \geq T'$, and therefore

$$
\begin{aligned}
\mathbf{Pr}_{X \in_u G}[|p(X)| \geq T] &\leq \mathbf{Pr}_{X \in_u G}[|p'(X)| \geq T'] \\
&\leq 2 \exp(-T'/3) + \varepsilon^2/(2T' \ln(T')^2) \\
&\leq 2 \exp(-(T - 2\varepsilon)/3) + \varepsilon^2/(2(T - 2\varepsilon) \ln^2(T - 2\varepsilon)) \\
&\leq 3 \exp(-T/4) + \varepsilon^2/(T \ln^2 T) \, ,
\end{aligned}
$$

566 where the second line follows from Claim A.6(b) for $(p', T')$. This completes the proof of Condition
567 (iv)(b).

568 The proof of Lemma A.3 is now complete. $\qquad\square$

569 Our algorithm iteratively applies the procedure ROBUST-SPARSE-MEAN (Algorithm 1). The crux
570 of the proof is the following performance guarantee of ROBUST-SPARSE-MEAN:

571 **Proposition A.7.** *Algorithm 1 has the following performance guarantee: On input a multiset $S$ of*
572 *$N$ points in $\mathbb{R}^d$ such that $\Delta(G, S) \leq 2\varepsilon$, where $G$ is an $(\varepsilon, k, \tau)$-good set with respect to $\mathcal{N}(\mu, I)$,*
573 *procedure ROBUST-SPARSE-MEAN returns one of the following:*

574      *1. A mean vector $\widehat{\mu}$ such that $\|\widehat{\mu} - \mu\|_2 = O(\varepsilon \sqrt{\log(1/\varepsilon)})$, or*

575      *2. A multiset $S' \subset S$ satisfying $\Delta(G, S') \leq \Delta(G, S) - \varepsilon/N$.*

576 We note that our overall algorithm terminates after at most $2N$ iterations of Algorithm 1, in which
577 case it returns a candidate mean vector satisfying the first condition of Proposition A.7. Note that
578 the initial $\varepsilon$-corrupted set $S$ satisfies $\Delta(G, S) \leq 2\varepsilon$. If $S^{(i)} \subset S$ is the multiset returned after the
579 $i$-th iteration, then we have that $0 \leq \Delta(S^{(i)}, G) \leq 2\varepsilon - i(\varepsilon/N)$.

580 In the rest of this section, we prove Proposition A.7.

581 We start by showing the first part of Proposition A.7. Note that Algorithm 1 outputs a candidate
582 mean vector only if $\|(\widetilde{\Sigma} - I)_{(U)}\|_F \leq O(\varepsilon \log(1/\varepsilon))$. We start with the following lemma:

583 **Lemma A.8.** *If $\|(\widetilde{\Sigma} - I)_{(U)}\|_F \leq O(\varepsilon \log(1/\varepsilon))$, then for any $T \subseteq [d]$ with $|T| \leq k$, we have*
584 *$\|\widetilde{\mu}_T - \mu_T\|_2 \leq O(\varepsilon \sqrt{\log(1/\varepsilon)})$.*

585 *Proof.* Fix $T \subseteq [d]$ with $|T| \leq k$. By definition, $\|(\widetilde{\Sigma} - I)_T\|_F$ is the Frobenius norm of the
586 corresponding sub-matrix on $T \times T$. Note that this is the $\ell_2$-norm of a set of $k$ diagonal entries and
587 $k^2 - k$ off-diagonal entries of $\widetilde{\Sigma} - I$. By construction, $U$ is the set that maximizes this norm, and
588 therefore

$$
\|(\widetilde{\Sigma} - I)_T\|_2 \leq \|(\widetilde{\Sigma} - I)_T\|_F \leq \|(\widetilde{\Sigma} - I)_{(U)}\|_F \leq O(\varepsilon \log(1/\varepsilon)) \, .
$$

589 Given this bound, we leverage a proof technique from [DKK+16] showing that a bound on the
590 spectral norm of the covariance implies a $\ell_2$-error bound on the mean. This implication is not
591 explicitly stated in [DKK+16], but follows directly from the arguments in Section 5.1.2 of that work.
592 In particular, the analysis of the "small spectral norm" case in that section shows that $\|\widetilde{\mu}_T - \mu_T\|_2 \leq$
593 $O\left(\sqrt{\varepsilon}\|(\widetilde{\Sigma} - I)_T\|_2^{1/2} + \varepsilon \sqrt{\log(1/\varepsilon)}\right)$, from which the desired claim follows. This completes the
594 proof of Lemma A.8. $\qquad\square$

595 Given Lemma A.8, the correctness of the sparse mean approximation output in Step 4 of Algorithm 1
596 follows from the following corollary:

597 **Corollary A.9.** *Let $\widehat{\mu} = h_k(\widetilde{\mu})$. If $\|(\widetilde{\Sigma} - I)_{(U)}\|_F \leq O(\varepsilon \log(1/\varepsilon))$, then $\|\widehat{\mu} - \mu\|_2 \leq$*
598 *$O(\varepsilon \sqrt{\log(1/\varepsilon)})$.*

599 *Proof.* For vectors $x, y$, let $N_x$ denote the set of coordinates on which $x$ is non-zero and $N_{x|y}$
600 denote the set of coordinates on which $x$ is non-zero and $y$ is zero. Setting $T = N_\mu$ and $T = N_{\widehat{\mu}|\mu}$
601 in Lemma A.8, we get that $\|\widetilde{\mu}_{N_\mu} - \mu\|_2 \leq O(\varepsilon \sqrt{\log(1/\varepsilon)})$ and $\|\widetilde{\mu}_{N_{\widehat{\mu}|\mu}}\|_2 \leq O(\varepsilon \sqrt{\log(1/\varepsilon)})$.

602  If $\widetilde{\mu}$ has $k$ or fewer non-zero coordinates, then $\widehat{\mu} = \widetilde{\mu}$ and $\|\widehat{\mu} - \mu\|_2 = \|\widetilde{\mu}_{N_\mu \cup N_{\widehat{\mu}|\mu}} - \mu\|_2 \leq$
603  $O(\varepsilon\sqrt{\log(1/\varepsilon)})$ and we are done. Otherwise, $\widehat{\mu}$ has exactly $k$ non-zero coordinates and so $|N_{\mu|\widehat{\mu}}| \leq$
604  $|N_{\widehat{\mu}|\mu}|$. Since the nonzero coordinates of $\widehat{\mu}$ are the $k$ largest magnitude coordinates of $\widetilde{\mu}$, for any
605  $i \in N_{\mu|\widehat{\mu}}$ and $j \in N_{\widehat{\mu}|\mu}$, we have that $|\widetilde{\mu}_i| \leq |\widetilde{\mu}_j|$. Since $\|\widetilde{\mu}_{N_{\widehat{\mu}|\mu}}\|_2 \leq O(\varepsilon\sqrt{\log(1/\varepsilon)})$, at least one
606  coordinate $j \in N_{\widehat{\mu}|\mu}$ must have $\widetilde{\mu}_j^2 \leq O(\varepsilon^2 \log(1/\varepsilon))/|N_{\widehat{\mu}|\mu}|$. Therefore, for any $i \in N_{\mu|\widehat{\mu}}$, we have
607  that $\widetilde{\mu}_i^2 \leq O(\varepsilon^2 \log(1/\varepsilon))/|N_{\widehat{\mu}|\mu}|$.

608  Thus, we have

$$\|\widetilde{\mu}_{N_{\mu|\widehat{\mu}}}\|_2^2 = \sum_{i \in N_{\mu|\widehat{\mu}}} \widetilde{\mu}_i^2 \leq \frac{|N_{\mu|\widehat{\mu}}| \cdot O(\varepsilon^2 \log(1/\varepsilon))}{|N_{\widehat{\mu}|\mu}|} \leq O(\varepsilon^2 \log(1/\varepsilon)) \,,$$

609  where the second inequality used that $|N_{\mu|\widehat{\mu}}| \leq |N_{\widehat{\mu}|\mu}|$.

610  Since $\|\widetilde{\mu}_{N_\mu} - \mu\|_2 \leq O(\varepsilon\sqrt{\log(1/\varepsilon)})$, by the triangle inequality we have that $\|\mu_{N_{\mu|\widehat{\mu}}}\|_2 \leq$
611  $O(\varepsilon\sqrt{\log(1/\varepsilon)})$. Finally, we have that

$$\|\mu - \widehat{\mu}\|_2^2 = \|\mu_{N_\mu \cap N_{\widehat{\mu}}} - \widehat{\mu}_{N_\mu \cap N_{\widehat{\mu}}}\|_2^2 + \|\mu_{N_{\mu|\widehat{\mu}}}\|_2^2 + \|\widetilde{\mu}_{N_{\widehat{\mu}|\mu}}\|_2^2 \leq O(\varepsilon^2 \log(1/\varepsilon)) \,,$$

612  concluding the proof.      $\square$

613  Lemma A.8 and Corollary A.9 give the first part of Proposition A.7.

614  We now analyze the complementary case that $\|(\widetilde{\Sigma} - I)_{(U)}\|_F = \Omega(\varepsilon \log(1/\varepsilon))$. In this case, we
615  apply two different filters, a linear filter (Steps 5-8), and a quadratic filter (Steps 9-11). To prove
616  the second part of Proposition A.7, we will show that at least one of these two filters: (i) removes at
617  least one point, and (ii) it removes more corrupted than uncorrupted points.

618  The analysis in the case of the linear filter follows by a reduction to the linear filter in [DKK$^+$16]
619  for the non-sparse setting (see Proposition 5.5 in Section 5.1 of that work). More specifically, the
620  linear filter in Steps 5-8 is essentially identical to the linear filter of [DKK$^+$16] restricted to the
621  $2k^2 \times 2k^2$ matrix $\widetilde{\Sigma}_{U'}$. We note that Definition A.2 implies that every restriction to $2k^2$ coordinates
622  satisfies the properties of the good set in the sense of [DKK$^+$16] (Definition 5.2(i)-(ii) of that work).
623  This implies that the analysis of the linear filter from [DKK$^+$16] holds in our case, establishing the
624  desired properties. Since the linear filter removes more corrupted points than uncorrupted points, it
625  will remove at most a $2\varepsilon$ fraction of the points over all the iterations.

626  If the condition of the linear filter does not apply, i.e., if $\|(\widetilde{\Sigma} - I)_{U'}\|_2 \leq O(\varepsilon \log(1/\varepsilon))$, the
627  aforementioned analysis in [DKK$^+$16] implies $\|\widetilde{\mu}_{U'} - \mu_{U'}\|_2 \leq O(\varepsilon\sqrt{\log(1/\varepsilon)})$. In this case, we
628  show that the second filter behaves appropriately.

629  Let $p(x)$ be the polynomial considered in the quadratic filter. We start with the following technical
630  lemma analyzing the expectation and variance of $p(x)$ under various distributions:

631  **Lemma A.10.** *The following hold true:*

632     *(i) For $Y \sim \mathcal{N}(\widetilde{\mu}, I)$, we have that $\mathbf{E}[p(Y)] = 0$ and $\mathbf{Var}[p(Y)] = 1$.*

633     *(ii) For $X \in_u S$, we have that $\mathbf{E}[p(X)] = \|(\widetilde{\Sigma} - I)_{(U)}\|_F$.*

634     *(iii) For $Z \sim \mathcal{N}(\mu, I)$, we have that $|\mathbf{E}[p(Z)]| \leq O(\varepsilon^2 \log(1/\varepsilon))$ and $\mathbf{Var}[p(Z)] = 1 +$*
635     *$O(\varepsilon^2 \log(1/\varepsilon))$.*

636  *Proof.* Let $A = \frac{(\widetilde{\Sigma} - I)}{\|(\widetilde{\Sigma} - I)\|_F}$ and $p(x) := (x - \widetilde{\mu})^T A_{(U)}(x - \widetilde{\mu}) - \text{Tr}[A_{(U)}]$. We have

$$\mathbf{E}[(Y - \widetilde{\mu})^T A_{(U)}(Y - \widetilde{\mu})] = \text{Tr}[A_{(U)}\mathbf{E}[(Y - \widetilde{\mu})(Y - \widetilde{\mu})^T]] = \text{Tr}[A_{(U)}I] = \text{Tr}[A_{(U)}] \,.$$

637  Therefore, $\mathbf{E}[p(Y)] = \text{Tr}[A_{(U)}] - \text{Tr}[A_{(U)}] = 0$. Similarly,

$$\mathbf{E}[(X - \widetilde{\mu})^T A_{(U)}(X - \widetilde{\mu})] = \mathbf{E}[\text{Tr}[A_{(U)}(X - \widetilde{\mu})(X - \widetilde{\mu})^T]] = \text{Tr}[A_{(U)}\mathbf{E}[(X - \widetilde{\mu})(X - \widetilde{\mu})^T]] = \text{Tr}[A_{(U)}\widetilde{\Sigma}] \,,$$

638 and so

$$\mathbf{E}[p(X)] = \mathrm{Tr}[A_{(U)}(\widetilde{\Sigma} - I)] = \mathrm{Tr}[A_{(U)}A]\|(\widetilde{\Sigma} - I)_{(U)}\|_F = \|A_U\|_F \|(\widetilde{\Sigma} - I)_{(U)}\|_F = \|(\widetilde{\Sigma} - I)_{(U)}\|_F \ .$$

639 We have thus shown (ii) and the first part of (i).

640 We now proceed to show the first part of (iii). Note that

$$\mathbf{E}[(Z - \widetilde{\mu})(Z - \widetilde{\mu})^T] = \mathbf{E}[(Z - \mu)(Z - \mu)^T] + \mathbf{E}[(\widetilde{\mu} - \mu)(Z - \widetilde{\mu})^T] + \mathbf{E}[(Z - \mu)(\widetilde{\mu} - \mu)^T]$$
$$= I + (\widetilde{\mu} - \mu)(\widetilde{\mu} - \mu)^T + 0.$$

641 Thus, we can write

$$\mathbf{E}[p(Z)] = \mathrm{Tr}[A_{(U)}(\mathbf{E}[(Z - \widetilde{\mu})(Z - \widetilde{\mu})^T] - I)] = (\widetilde{\mu} - \mu)^T A_{(U)}(\widetilde{\mu} - \mu) \ ,$$

642 and so

$$|\mathbf{E}[p(Z)]| \leq \|\widetilde{\mu}_{U'} - \mu_{U'}\|_2^2 \|A_{(U)}\|_2 \leq \|\widetilde{\mu}_{U'} - \mu_{U'}\|_2^2 \leq O(\varepsilon^2 \log(1/\varepsilon)) \ .$$

643 This proves all the statements about expectations.

644 We now analyze the variance of $p(x)$ for $Y$ and $Z$. Since $A_{(U)}$ is symmetric, we can write $A_{(U)} =$
645 $O^T \Lambda O$ for an orthogonal matrix $O$ and a diagonal matrix $\Lambda$. Note that $Y' = O(Y - \widetilde{\mu})$ is distributed
646 as $\mathcal{N}(0, I)$. Under these substitutions, $p(Y) = \sum_i \Lambda_{ii} Y_i'^2$, and so

$$\mathbf{Var}[p(Y)] = \sum_i \Lambda_{ii}^2 \mathbf{Var}[Y_i'^2] = \|\Lambda\|_F^2 = \|A_{(U)}\|_F^2 = 2.$$

647 Similarly, if we take $Z' = O^T(Z - \widetilde{\mu})$, then $Z' \sim \mathcal{N}(O^T(\widetilde{\mu} - \mu), I)$. We have $\mathbf{E}[Z_i'] = (O^T(\widetilde{\mu} - \mu))_i$
648 and, letting $Z'' = Z' - \mathbf{E}[Z']$, we get that

$$\mathbf{E}[Z_i'^2] = \mathbf{E}[(Z_i'' + \mathbf{E}[Z_i'])^2] = \mathbf{E}[Z_i''^2] + 2\mathbf{E}[Z_i']\mathbf{E}[Z_i''] + \mathbf{E}[Z_i']^2 = 1 + 0 + \mathbf{E}[Z_i']^2 \ .$$

649 Next we can write

$$\mathbf{E}[Z_i'^4] = \mathbf{E}[(Z_i'' + \mathbf{E}[Z_i'])^4] = \mathbf{E}[Z_i''^4] + 0 + 6\mathbf{E}[Z_i''^2]\mathbf{E}[Z_i']^2 + \mathbf{E}[Z_i']^4 = 2 + 6\mathbf{E}[Z_i']^2 + \mathbf{E}[Z_i']^4 \ .$$

650 Thus, $\mathbf{Var}[Z_i'^2] = 2 + 6\mathbf{E}[Z_i']^2 + \mathbf{E}[Z_i']^4 - (1 + \mathbf{E}[Z_i']^2)^2 = 1 + 4\mathbf{E}[Z_i']^2$. We therefore have

$$\mathbf{Var}[p(Z)] = \sum_i \Lambda_{ii}^2 \mathbf{Var}[Z_i'^2] = \sum_i \Lambda_{ii}^2(1 + 4\mathbf{E}[Z_i']^2) = \sum_i \Lambda_{ii}^2 + 4\Lambda_{ii}^2(O^T(\widetilde{\mu} - \mu))_i$$
$$= \|\Lambda\|_F^2 + (\widetilde{\mu} - \mu)^T O\Lambda^2 O^T(\widetilde{\mu} - \mu) = 1 + (\widetilde{\mu} - \mu)^T A_{(U)}^2(\widetilde{\mu} - \mu)$$
$$\leq 1 + \|\widetilde{\mu}_{U'} - \mu_{U'}\|_2^2 \cdot \|A_{(U)}^2\|_2^2 \leq 1 + O(\varepsilon^2 \log(1/\varepsilon)) \cdot 1 \ .$$

651 This completes the proof of Lemma A.10. $\qquad\square$

652 Suppose that we find a threshold $T > 0$ such that Step 10 of the algorithm holds, i.e., the quadratic
653 filter applies. Then we can show that Step 11 removes more bad points than good points. This
654 follows from standard arguments, by combining Definition A.2(iv)(b) with our upper bound for
655 $\mathbf{E}_{Z \sim \mathcal{N}(\mu, I)}[p(Z)]$ from Lemma A.10. Let $S = G \cup E \setminus L$. By Definition A.2(iv)(b), for the
656 good set $G$, we have that for $X \in_u G$, $\mathbf{Pr}\left[\left|p(X) - \mathbf{E}_{Z \sim \mathcal{N}(\mu, I)}[p(Z)]\right| \geq T\right] \leq 3\exp(-T/4) +$
657 $\varepsilon^2/(T \ln^2 T)$. Lemma A.10(iii) implies that $|\mathbf{E}[p(Z)]| \leq O(\varepsilon^2 \log(1/\varepsilon))$. Therefore, we obtain the
658 following corollary:

659 **Corollary A.11.** *We have that:*

660     *(i)* $|\mathbf{E}_{X \in_u G}[p(X)]| \leq O(\varepsilon)$ *and,*

661     *(ii) For* $T \geq 6$, $\mathbf{Pr}_{X \in_u G}[|p(X)| \geq T] \leq (3 + O(\varepsilon))\exp(-T/4) + (1 + O(\varepsilon))\left(\varepsilon^2/(T \ln^2 T)\right)$.

662 Condition (ii) implies that the fraction of points in $G$ that violate the quadratic filter condition is less
663 than $1/2$ the fraction of points in $S$ that violate the same condition. Therefore, the quadratic filter
664 removes more bad points than good points.

665 It remains to show that if Algorithm 1 does not terminate in Step 4 and the linear filter does not
666 apply, then the quadratic filter necessarily applies. To establish this, we need a couple more technical
667 lemmas. We first show that the expectation of $p(x)$ over the set of good samples that are removed is
668 small:

**Lemma A.12.** *We have that* $|L| \cdot |\mathbf{E}_{X \in_u L}[p(X)]| \leq |S| \cdot O(\varepsilon \log(1/\varepsilon))$.

*Proof.* Since $L \subset G$ and $|G| = O(|S|)$, for $T \geq 6$ we have

$$|L| \cdot \mathbf{Pr}_{X \in_u L}[|p(X)| \geq T] \leq |G| \cdot \mathbf{Pr}_{X \in_u G}[|p(X)| \geq T] \leq O\left(|S|(\exp(-T/4) + \varepsilon^2/(T \ln^2 T))\right) ,$$

where we used Corollary A.11. Thus, we obtain that

$$
\begin{aligned}
|L| \cdot |\mathbf{E}_{X \in_u L}[p(X)]| &\leq |L| \cdot \mathbf{E}_{X \in_u L}[|p(X)|] \\
&= \int_0^\infty |L| \cdot \mathbf{Pr}_{X \in_u L}[|p(X)| \geq T] dT \\
&\leq \int_0^{3\ln(1/\varepsilon)} |L| dT + \int_{3\ln(1/\varepsilon)}^\infty O(|S|(\exp(-T/4) + \varepsilon^2/(T \ln^2 T))) dT \\
&\leq O(|S|\varepsilon \log(1/\varepsilon)) + O(|S|\varepsilon) + O(|S|\varepsilon^2/\log\log(1/\varepsilon)) \\
&= O(|S|\varepsilon \log(1/\varepsilon)) ,
\end{aligned}
$$

where we used the fact that $|L| = O(\varepsilon|S|)$ and that the derivative of $1/\ln x$ is $1/x \ln^2 x$. This completes the proof of Lemma A.12. $\qquad\square$

By a similar argument, we can show that if the quadratic filter does not apply, then the remaining points in $E$ contribute a small amount to the expectation of $p(x)$.

**Lemma A.13.** *Suppose that for all* $T \geq 6$, *we have* $\mathbf{Pr}_{X \in_u S}[|p(X)| \geq T] \leq 9\exp(-T/4) + 3\varepsilon^2/(T \ln^2 T)$. *Then, we have that* $|E| \cdot |\mathbf{E}_{X \in_u E}[p(X)]| \leq O(|S|\varepsilon \log(1/\varepsilon))$.

By combining the above, we obtain the following corollary, completing the analysis of our algorithm:

**Corollary A.14.** *If we reach Step 10 of Algorithm 1, then there exists a* $T \geq 6$ *such that* $\mathbf{Pr}_{X \in_u S}[|p(X)| \geq T] \geq 9\exp(-T/4) + 3\varepsilon^2/(T \ln^2 T)$.

*Proof.* Suppose for a contradiction that no such $T$ exists. Using Corollary A.11, Lemmas A.12 and A.13, we obtain that

$$
\begin{aligned}
|S| \cdot \|(\widetilde{\Sigma} - I)_U\|_F = |S| \cdot \mathbf{E}_{X \in_u S}[p(X)] &= |G| \cdot \mathbf{E}_{X \in_u G}[p(X)] + |E| \cdot \mathbf{E}_{X \in_u E}[p(X)] - |L| \cdot \mathbf{E}_{X \in_u L}[p(X)] \\
&= O(|S|\varepsilon \log(1/\varepsilon)) .
\end{aligned}
$$

This is a contradiction, as if this was the case, Algorithm 1 would have retuned in Step 4. $\qquad\square$

# B  Robust Sparse PCA

In this section, we prove correctness of Algorithm 2 establishing Theorem 1.3, which we restate for completeness:

**Theorem B.1.** *Let* $D \sim \mathcal{N}(0, I + \rho vv^T)$ *be a centered Gaussian distribution on* $\mathbb{R}^d$ *with spiked covariance* $\Sigma = I + \rho vv^T$ *for an unknown $k$-sparse unit vector $v$, and $0 < \rho < O(1)$ a real number. For some $\varepsilon > 0$, let $S$ be an $\varepsilon$-corrupted set of samples from $D$ of size $N = \Omega(k^4 \log^4(d/\varepsilon)/\varepsilon^2)$. There exists an algorithm that, on input $S$, $k$, and $\varepsilon$, runs in polynomial time and returns $w \in \mathbb{R}^d$ such that with probability at least $2/3$ we have that $\|ww^T - vv^T\|_F = O\left(\frac{\varepsilon \log(1/\varepsilon)}{\rho}\right)$.*

*We will require some additional notation. For any $M \in \mathbb{R}^{d \times d}$, define $\mathsf{vec}(M) \in \mathbb{R}^{d^2}$ to be a canonical flattening of this vector, $\gamma(x) \in \mathbb{R}^{d^2}$ to be $\mathsf{vec}(xx^T - I)$.*

As is standard with such robust statistics arguments, we will need to assume that the uncorrupted set of good samples $G$ has some desired properties. In particular, we will make use of the following notion of a good set:

**Definition B.2.** *Define a set $G \subset \mathbb{R}^n$ to be $(\varepsilon, k)$-good for $\mathcal{N}(0, I + \rho vv^T)$ and $\rho > 0$ if the following hold for every $Q \subset [d] \times [d]$*

1. *For some sufficiently large constant $C$ and for every $i \in [d]$ and $x \in G$, $|x_i| \leq C\sqrt{\log(d|G|)}$.*

2. $\left\| (\mathbf{E}_G[xx^T] - I - \rho vv^T)_Q \right\|_F \leq \varepsilon$

3. *For all $w \in \mathbb{R}^{k^2}$,*

$$\mathbf{Var}_G(\gamma(x)_Q \cdot w) = (1 \pm \varepsilon)\mathbf{Var}_{\mathcal{N}(0, I + \rho vv^T)}(\gamma(x)_Q \cdot w)$$

4. *For $C$ a sufficiently large constant, and for all $w \in \mathbb{R}^{k^2}$ satisfying $\|w\|_2 = 1$, and all $T > \log(1/\varepsilon)$*

$$\mathbf{Pr}_G[|\gamma(x)_Q \cdot w - \rho\mathsf{vec}_Q(vv^T) \cdot w| > CT] < \frac{\varepsilon}{T^2 \log^2(T)}$$

We note that given a sufficiently large set of independent samples from $X$ that the above conditions hold with high probability.

**Lemma B.3.** *If $G$ is a set of $N = Ck^4 \log^4(d/\varepsilon)/\varepsilon^2$ samples drawn from $\mathcal{N}(0, I + \rho vv^T)$, for $C$ a sufficiently large constant. Then $G$ is $(\varepsilon, k)$-good with probability at least $2/3$.*

*Proof.* Condition 1 follows from standard gaussian concentration bounds. To see that Condition 2 holds, we prove entrywise closeness of the matrices involved. We will use the following standard concentration inequality

**Lemma B.4.** *Any degree $d$ polynomial $f(A_1, \ldots, A_n)$ of independent centered Gaussian random variables $A_1, \ldots, A_n$ satisfies*

$$\mathbf{Pr}\left(|f(A) - \mathbb{E}[f(A)]| > \tau\right) \lesssim e^{-\left(\frac{\tau^2}{R \cdot \mathrm{Var}(f(A))}\right)^{1/d}}$$

*where $R$ is some universal constant.*

Entries of $xx^T - (I + \rho vv^T)$ are degree 2 polynomials of Gaussians, and thus so is their mean over $G$. Hence, Lemma B.4 implies that for any $(i, j) \in [d] \times [d]$ that

$$\mathbf{Pr}\left(|\mathbf{E}_G[x_i x_j] - (\delta_{i,j} + \rho v_i v_j)| > \varepsilon/k\right) \lesssim \exp\left(-(N\varepsilon^2/Rk^2)^{1/2}\right).$$

Taking a union bound over $i, j$ shows that with high probability $\mathbf{E}_G[xx^T]$ has each entry within $\varepsilon/k$ of that of $\rho vv^T + I$, and this immediately implies Condition 2.

Condition 3 holds via a similar argument. Observe that it is sufficient to consider the case $\|w\|_2 = 1$ and sample enough points to satisfy

$$|\mathbf{E}_G[(x_i x_j - \delta_{i,j} - \rho v_i v_j)(x_k x_l - \delta_{k,l} - \rho v_k v_l)] - \mathbf{E}_{\mathcal{N}(0, I + \rho vv^T)}[(x_i x_j - \delta_{i,j} - \rho v_i v_j)(x_k x_l - \delta_{k,l} - \rho v_k v_l)]| \leq \frac{\varepsilon}{k^2}.$$

Then the spectral norm of the covariance matrix of $\gamma(x)$ for any $Q \times Q$ submatrix will also be bounded by $\varepsilon$. Note that this is just the probability that a degree-4 polynomial in Gaussian inputs deviates too much from its mean, and thus by Lemma B.4 the probability that the above fails to hold for any $(i, j, k, l)$ is at most

$$\exp\left(-(N\varepsilon^2/Rk^4)^{1/4}\right).$$

Taking a union bound over $(i, j, k, l)$ yields our result.

Finally, for Condition 4, we note that (perhaps changing the constant $C$), it suffices to prove it for all $\binom{d^2}{k}$ possible $Q$'s and for all $w$ in a cover of the unit ball of $\mathbb{R}^{k^2}$ (which will have size $2^{O(k^2)}$ and for $T$ powers of 2 less than or equal to $k \log(dN)$ (since by Condition 1 $|\mathsf{vec}_Q(xx^T)| = O(k \log(dN))$ for all $x \in G$). Once we have fixed $Q, w$ and $T$, $\gamma_Q(x) \cdot w - \rho\mathsf{vec}_Q(vv^T) \cdot w$ is a mean 0, variance

731  $O(1)$, degree-2 polynomial so by Lemma B.4, the probability that it is more than $CT$ is at most
732  $e^{-2T}$. Then the probability that at least $\varepsilon N/(T^2 \log^2(T))$ of our $x$'s have this property is at most

$$\binom{N}{\varepsilon N/(T^2 \log^2(T))} \exp(-2T(\varepsilon N)/(T^2 \log^2(T))) \le \left(\frac{Ne^{-2T}}{e\varepsilon N/(T^2 \log^2(T))}\right)^{(\varepsilon N)/(T^2 \log^2(T))}$$
$$\le \exp(-\Omega(T\varepsilon N/(T^2 \log^2(T))))$$
$$\le \exp(-\Omega(\varepsilon N/T \log^2(T)))$$
$$\le \exp(-\Omega(k^3 \log(d/\varepsilon))).$$

733  Taking a union bound over $Q, w, T$ completes the proof.

734  $\qquad\qquad\qquad\qquad\qquad\qquad\qquad\qquad\qquad\qquad\qquad\qquad\qquad\qquad\qquad\qquad$ □

735  We think in fact that we should be able to produce a good set with substantially fewer samples.

736  **Conjecture 1.** *There exists an $N = k^2 \text{polylog}(d/\varepsilon)/\varepsilon^2$ so that if $G$ is a set of $N$ samples drawn*
737  *from $\mathcal{N}(0, I + \rho vv^T)$, then $G$ is $(\varepsilon, k)$-good with probability at least $1 - 1/d$.*

738  We can now proceed with the proof of our main Theorem. In particular, our algorithm will follow
739  quickly from the existence of the following subroutine:

740  **Proposition B.5.** *Let $G$ be an $(\varepsilon, k)$-good set for $\mathcal{N}(0, \Sigma)$ with $\Sigma = I + \rho vv^T$ with $v$ a unit*
741  *length, $k$-sparse vector and $0 < \rho < 1$. There exists an algorithm (Algorithm 2) that given a*
742  *matrix $\tilde{\Sigma}$ and a set $S$ with $\|\tilde{\Sigma} - \Sigma\|_F \le \delta$ and $\Delta(S, G) \le \varepsilon|G|$ returns either a matrix $\Sigma'$ with*
743  *$\|\Sigma' - \Sigma\|_F = O(\sqrt{\varepsilon\delta} + \varepsilon \log(1/\varepsilon))$ or a subset $T \subset S$ with $\Delta(T, G) < \Delta(S, G)$.*

744  Our main theorem follows from iteratively applying the Proposition. The error stabilizes at $\delta$ with
745  $\delta = O(\sqrt{\varepsilon\delta} + \varepsilon \log(1/\varepsilon))$, which implies that $\delta = O(\varepsilon \log(1/\varepsilon))$. We begin by analyzing what
746  happens when our algorithm returns a matrix. We first note that if we pass the filter, then $\tilde{\mu}_Q$ will be
747  approximately correct.

748  **Lemma B.6.** *With the notation as in Algorithm 2, we have that $\|\tilde{\mu}_Q - \text{vec}_Q(\Sigma - I)\|_2 = O(\sqrt{\varepsilon\lambda} +$*
749  *$\sqrt{\varepsilon\delta} + \varepsilon \log(1/\varepsilon))$.*

750  *Proof.* Let $\|\tilde{\mu}_Q - \text{vec}_Q(\Sigma - I)\|_2 = a$.

751  Let $S = (G\backslash L) \cup E$. We wish to show that

$$\left\|\sum_{x \in S}(xx^T - \Sigma)_Q\right\|_2 = O(\sqrt{\varepsilon\delta} + \varepsilon \log(1/\varepsilon))|G|.$$

752  We note that the left hand side above is at most

$$\left\|\sum_{x \in G}(xx^T - \Sigma)_Q\right\|_2 + \left\|\sum_{x \in L}(xx^T - \Sigma)_Q\right\|_2 + \left\|\sum_{x \in E}(xx^T - \Sigma)_Q\right\|_2.$$

753  Since $G$ is a good set, by Condition 2, we have that the first term is $O(\varepsilon|G|)$. The second term is at
754  most the supremum over unit vectors $w \in \mathbb{R}^{k^2}$ of

$$\sum_{x \in L}(w \cdot \gamma_Q(x) - \rho w \cdot \text{vec}_Q(vv^T))$$

755  This is at most

$$\int_0^\infty \left|\{x \in L : |(w \cdot \gamma_Q(x) - \rho w \cdot \text{vec}_Q(vv^T)| > t\}\right| dt.$$

756  When $U = L$, this is at most

$$\int_0^{C \log(1/\varepsilon)} \varepsilon|G| + \int_{C \log(1/\varepsilon)}^\infty \varepsilon/((t/C)^2 \log^2(t/C))|G| dt = O(\varepsilon \log(1/\varepsilon)|G|),$$

757 where the bound on the second term above is by Condition 4.

758 We can bound the final term by Cauchy-Schwartz as

$$(\varepsilon|G|)^{1/2} \left( \sum_{x \in E} (w \cdot (xx^T - \Sigma)_Q)^2 \right)^{1/2}.$$

759 To bound this we note that

$$\sum_{x \in S} (w \cdot (xx^T - \Sigma)_Q)^2 \leq |S|(\mathbf{Var}_S(w \cdot \gamma_Q(x)) + a^2).$$

760 Now we know that

$$\mathbf{Var}_G(w \cdot \gamma_Q(x)) = \mathbf{Var}_{\mathcal{N}(0, \rho vv^T + I)}(w \cdot \gamma_Q(x)) + O(\varepsilon) = \mathbf{Var}_{\mathcal{N}(0, \widetilde{\Sigma})}(w \cdot \gamma_Q(x)) + O(\varepsilon + \delta).$$

761 Thus,

$$\sum_{x \in G} ((w \cdot (xx^T - \Sigma)_Q)^2 - \mathbf{Var}_{\mathcal{N}(0, \widetilde{\Sigma})}(w \cdot \gamma_Q(x))) = O(\varepsilon + \delta)|G|.$$

762 However, since $v^*$ is a maximum eigenvalue, we also have that

$$\mathbf{Var}_S(w \cdot \gamma_Q(x)) - \mathbf{Var}_{\mathcal{N}(0, \widetilde{\Sigma})}(w \cdot \gamma_Q(x)) \leq \lambda + a^2.$$

763 Combining with the above, we have that

$$\sum_{x \in E} ((w \cdot (xx^T - \Sigma)_Q)^2 - \mathbf{Var}_{\mathcal{N}(0, \widetilde{\Sigma})}(w \cdot \gamma_Q(x))) \leq \sum_{x \in L} ((w \cdot (xx^T - \Sigma)_Q)^2 - \mathbf{Var}_{\mathcal{N}(0, \widetilde{\Sigma})}(w \cdot \gamma_Q(x)))$$
$$+ |G|O(\varepsilon + \delta + \lambda + a^2).$$

764 However, we can bound

$$\sum_{x \in L} ((w \cdot (xx^T - \Sigma)_Q)^2 - \mathbf{Var}_{\mathcal{N}(0, \widetilde{\Sigma})}(w \cdot \gamma_Q(x)))$$

765 by

$$O(1) + \int_0^\infty \left| \{x \in L : |(w \cdot \gamma_Q(x) - \rho w \cdot \mathsf{vec}_Q(vv^T)| > t\} \right| 2tdt.$$

766 As before, this is at most

$$\int_0^{C \log(1/\varepsilon)} 2t\varepsilon|G|dt + \int_{C \log(1/\varepsilon)}^\infty \varepsilon/((t/C)^2 \log^2(t/C))|G|2tdt = O(\varepsilon \log^2(1/\varepsilon)|G|).$$

767 Thus, the final term in our sum is at most

$$(\varepsilon|G|)^{1/2} O(\varepsilon \log^2(1/\varepsilon)|G| + (a^2 + \varepsilon + \delta + \lambda)|G|)^{1/2}$$

768 Therefore, we have that

$$a = O(a\sqrt{\varepsilon} + \sqrt{\varepsilon\lambda} + \sqrt{\varepsilon\delta} + \varepsilon \log(1/\varepsilon)),$$

769 from which we conclude our result.

770 $\qquad\qquad\qquad\qquad\qquad\qquad\qquad\qquad\qquad\qquad\qquad\qquad\qquad\qquad\qquad\qquad\qquad\qquad\quad$ $\square$

771 Given this, we would like to show that $\Sigma'$ is close to $\Sigma$. In particular, we have:

772 **Lemma B.7.** *Suppose that* $A = \mathbf{E}_{x \in_u S}[xx^T - I]$ *and $Q$ the set of its $k^2$ largest entries. If* $\|(A -$
773 $\rho vv^T)_Q\|_F = \eta$ *then for $w$ a normalized, principle eigenvector of $A_Q$ we have that $\rho w$ is within*
774 $O(\eta + \varepsilon \log(1/\varepsilon))$ *of either $\rho v$ or $-\rho v$.*

775 Before we begin with the proof, we make an important observation:

**Lemma B.8.** *In the notation above, for any set of entries $R$ defining a $k^2 \times k^2$ submatrix, $(A+I)_R \geq ((\rho v v^T + I) - O(\varepsilon \log(1/\varepsilon))I)_R$, as self-adjoint operators.*

*Proof.* Note that $A + I = \mathbf{E}_{x \in_u S}[xx^T] \geq (1-\varepsilon)\mathbf{E}_{x \in_u G \setminus L}[xx^T]$. By Property 2 of what it means to be a good set, $\mathbf{E}_{x \in_u G}[xx^T] = \rho v v^T + I + O(\varepsilon)$. Thus, it suffices to show that for any unit vector $u$ with support of size at most $k^2$ that $|L|/|G|\mathbf{E}_{x \in_u L}[(x \cdot u)^2] = O(\varepsilon \log(1/\varepsilon))$. This follows easily from Property 4. $\square$

We are now ready to prove Lemma B.7.

*Proof.* Let $R$ be the support of $vv^T$. Note that $A$ has larger total $L^2$ mass on $Q$ than it does on $R$. Therefore,

$$\|A_{R \setminus Q}\|_F \leq \|A_{Q \setminus R}\|_F \leq \|(A - \rho v v^T)_Q\|_F = \eta.$$

Let $B = (\rho v v^T)_{R \setminus Q}$. We note that with respect to Frobenius norm:

$$A_Q = A_{Q \cap R} + O(\eta) = (\rho v v^T)_{Q \cap R} + O(\eta) = (\rho v v^T - B) + O(\eta).$$

We also note that this is $A_{Q \cap R} + O(\eta) = A_R + O(\eta)$. Combining this with the above lemma, we have that

$$(\rho v v^T - B + I) + O(\eta) \geq (\rho v v^T + I) - O(\varepsilon \log(1/\varepsilon))I.$$

Rearranging, we find that $B \leq O(\eta + \varepsilon \log(1/\varepsilon))I$. But we note that the sign of the $i,j$ entry of $B$ is the same as the sign of $v_i v_j$ or 0. This means that $B$ is similar to a matrix with non-negative entries, and thus by The Perron–Frobenius Theorem, the largest eigenvalue of $B$ is positive, and hence $\|B\|_2 = O(\eta + \varepsilon \log(1/\varepsilon))$. Therefore, we have that

$$\|A_Q - \rho v v^T\|_2 \leq \|A_Q - (\rho v v^T - B)\|_2 + \|B\|_2 = O(\eta + \varepsilon \log(1/\varepsilon)).$$

Note that unless $\varepsilon$ and $\eta$ are sufficiently small, there is nothing to prove. Otherwise, we have that $v \cdot A_Q v \geq \rho - O(\eta + \varepsilon \log(1/\varepsilon))$, so $w$ will be an eigenvector with some eigenvalue $\lambda > \rho/2$. Since $\|A_Q - \rho v v^T\|_2 < \rho/2$, this means that $w$ must have a non-trivial component in the $v$-direction. Assume that $w$ is proportional to $v + u$ with $u$ orthogonal to $v$. Then we have that

$$\lambda(v + u) = \lambda w = A_Q w = A_Q(v + u) = \rho v + O(\eta + \varepsilon \log(1/\varepsilon)).$$

Taking the perpendicular to $v$ component above, we have that $\|u\|_2 = O(\eta + \varepsilon \log(1/\varepsilon))$, and this completes our proof. $\square$

Finally, note that

$$\begin{aligned}
\|vv^T - ww^T\|_F^2 &= \|vv^T\|_F^2 + \|ww^T\|_F^2 - 2\mathrm{tr}(vv^T ww^T) \\
&= 2 - 2(v \cdot w)^2 \ll 2 - 2|v \cdot w| = \|v \pm w\|_2^2 \\
&= \frac{\|\rho v \pm \rho w\|_2^2}{\rho^2} \leq O\left(\left(\frac{\eta + \varepsilon \log(1/\varepsilon)}{\rho}\right)^2\right)
\end{aligned}$$

Thus, plugging in $\eta = O(\sqrt{\varepsilon \lambda} + \sqrt{\varepsilon \delta} + \varepsilon \log(1/\varepsilon))$ above, we find that $\|vv^T - ww^T\|_F = O(\frac{\sqrt{\varepsilon \delta} + \varepsilon \log(1/\varepsilon)}{\rho})$.

801  We have left to analyze what happens when our algorithm returns a set $S'$. It is easy to see by
802  Conditions 2 and 3 that only $1/3$ of the elements of $G$ have $(\gamma(x)_Q - \rho\mathsf{vec}_Q(vv^T)) \cdot v^* > 3$.
803  Therefore, we have that $\hat\mu$ is within 3 of $\rho v^* \cdot (\mathsf{vec}_Q(vv^T))$. From this and Condition 4 it is easy to
804  see that if $C$ is sufficiently large (even compared to the $C$ in Condition 4), that less than half of the
805  elements of $S$ with $|\mathsf{vec}(xx^T) \cdot v^*| > CT + 3$ will be in $G$, and thus $\Delta(S', G) < \Delta(S, G)$.

806  All that remains is to show that such a threshold $T$ exists. To do this consider

$$\mathbf{Var}_S(v^* \cdot \mathsf{vec}_Q(xx^T)) - \mathbf{Var}_{\mathcal{N}(0,\Sigma)}(v^* \cdot \mathsf{vec}_Q(xx^T)).$$

807  This is $O(\delta) + \lambda$. On the other hand

$$\mathbf{Var}_S(v^* \cdot \mathsf{vec}_Q(xx^T)) = \mathbf{E}_S[(v^* \cdot \mathsf{vec}_Q(xx^T - \rho vv^T))^2] - \mathbf{E}_S[v^* \cdot \mathsf{vec}_Q(xx^T - \rho vv^T)]^2$$
$$= \mathbf{E}_S[(v^* \cdot \mathsf{vec}_Q(xx^T - \rho vv^T))^2] + O(\varepsilon\lambda + \varepsilon\delta + \varepsilon^2 \log^2(1/\varepsilon))$$

808  by Lemma B.6. Thus,

$$\mathbf{E}_S[(v^* \cdot \mathsf{vec}_Q(xx^T - \rho vv^T))^2] \geq \mathbf{Var}_{\mathcal{N}(0,\Sigma)}(v^* \cdot \mathsf{vec}_Q(xx^T)) + \lambda/2.$$

809  Now by Conditions 2 and 3 we have that

$$\sum_{x \in G}((v^* \cdot \mathsf{vec}_Q(xx^T - \rho vv^T))^2 - \mathbf{Var}_{\mathcal{N}(0,\Sigma)}(v^* \cdot \mathsf{vec}_Q(xx^T))) = O(|G|\varepsilon).$$

810  By arguments from the proof of Lemma B.6, we also have that

$$\sum_{x \in L}((v^* \cdot \mathsf{vec}_Q(xx^T - \rho vv^T))^2 - \mathbf{Var}_{\mathcal{N}(0,\Sigma)}(v^* \cdot \mathsf{vec}_Q(xx^T))) = O(|G|\varepsilon \log^2(1/\varepsilon)).$$

811  Thus, we must have

$$\sum_{x \in E}(v^* \cdot \mathsf{vec}_Q(xx^T - \rho vv^T))^2 \gg |G|\lambda.$$

812  However, this is at most

$$O\left(|E| + \int_0^\infty \left|\{x \in E : |v^* \cdot \mathsf{vec}_Q(xx^T) - \hat\mu > CT + 3\}\right| t dt\right).$$

813  If there is no such threshold, this is at most

$$O\left(|E| + \int_0^{\log(1/\varepsilon)} |E| t dt + \int_{\log(1/\varepsilon)}^\infty \varepsilon/(t^2 \log^2(t)) t dt\right) = O(\varepsilon \log^2(1/\varepsilon)|G|),$$

814  which is a contradiction. This completes our proof.

## Footnotes

\*Recall that a degree-$d$ polynomial is called homogeneous if its non-zero terms are all of degree exactly $d$.