[Reviews · NeurIPS 2019]

Reviewer 1



The paper studies the problem of robust sparse Mean Estimation, and robust sparse PCA in the strong-adversary model, i.e. an adversary inspects your samples, and can replace a constant fraction of them. In this setting, the authors get near optimal bounds, along with near-optimal sample complexities, in particular, the sample complexity scales only logarithmically with the dimension p. The main benefit of this proposed algorithm is that it escapes doing sparse-eigenvector computations, which was a bottleneck in previous works. The authors achieve this by carefully exploiting the problem structure, namely, using the knowledge of true covariance matrix, to construct a simple test to identify outliers. The proposed algorithm only requires eigenvector computations, and hence is a practical algorithm. Moreover, the authors do a good job at testing out their proposed algorithm by conducting extensive simulations. The paper is clearly written, and the authors have done a good job at giving an intuitive explanation of their algorithm.

Reviewer 2



This paper revisits the setting of robust statistics, specifically the mean estimation and PCA problems for Gaussian distributions with adversarial errors. It proposes algorithms for estimation with improved performance in time and sample complexity when the solution is sparse. For the mean estimation problem, the sample complexity matches the information theoretic optimum, and the error rate matches that of the best efficient algorithms, which is conjectured to be optimal for efficient algorithms. For PCA, the sample complexity is roughly quadratically worse than the optimum. Both methods also feature a significantly reduced running time compared to a previous (ellipsoid based) algorithm, though this is still quadratic in the ambient dimension and number of points. For mean estimation, the algorithm of Cheng-Diakonikolas-Ge from this year's SODA seems to obtain a different tradeoff that could be better under some circumstances, i.e., when Nd is large compared to epsilon. The question is important and the theoretical guarantees are nice. I note that other works on robust mean estimation in particular have considered much more general distributions than the Gaussian setting considered here though. I also have some serious concerns about the presentation. In the first place, the algorithm itself contains a number of undefined terms and notation. In particular, what is the h_k function? Also, I can guess what makes a set of examples "good" but not the particular, parameterized notion you use. Second, I could not understand the sketch of the main argument for mean estimation, specifically around lines 167-170. I realize the NeurIPS page limit is harsh, but if I can't get an appreciation for the key new theoretical ideas from the paper, that is a problem. This would be more useful than the nop Conclusion section. The current presentation simply does not stand alone. The empirical evaluation of the sample complexity and error rates seems pretty thorough, and this is good. One thing I would really have liked to see that is missing is the running time for the new method; the improved sample complexity is perhaps the headline, but one of the real problems with the prior, ellipsoid-based algorithms is the running time, and so it would have been ideal to report this improvement. I am pretty confident that it should be possible to beat an ellipsoid-based algorithm in practice, but why not report those numbers? A comparison with Cheng-Diakonikolas-Ge would also be very informative. ============== The running time numbers are helpful and I do encourage including them; the comparison to the running time of the first SDP of the ellipsoid-based method for the small instance gives a good point of reference. Also, thank you for the comparison to Cheng et al. Regarding the presentation, the description in the rebuttal (and the response to reviewer #3) were helpful. As a general principle I would strongly *discourage* using notation in the body of your paper that is only defined in an Appendix (in this case, the supplementary material). If the paper is accepted, please clean this up.

Reviewer 3



This paper is applying the existing filtering idea to a new problem, so the bar should be raised higher. I think it has good ideas, but there are a few issues the authors should address in order to increase the competitiveness of this work. 1. Theorem 1.2 talks about sparse mean estimation for Gaussian with identity cov. However, if k = d, which means that we have no sparsity assumption, the sample complexity the authors get is d^2 log d, which is way worse than the optimal d. Why do the authors claim it is optimal? 2. In general, the filtering idea and the ellipsoid idea are very similar: the ellipsoid idea relies on an unknown parameter, and the filtering idea is trying new parameters again and again. Hence, it would be very important to explain how the authors' test statistic compare with the ellipsoid algorithm approach. Are they the same or similar? If so, the value of this paper is watered down significantly. I read the rebuttal, and decided to increase my score since the authors have clearly demonstrated what their concrete contributions are.

[Author Response · NeurIPS 2019]

Thanks to all of the reviewers for the time spent reading and commenting on our work.

**General Comments:** Our main contributions are efficient outlier-robust algorithms for sparse recovery problems
(sparse mean estimation and sparse PCA) that rely on a novel spectral filtering method. Previous polynomial time
algorithms for these problems inherently relied on convex optimization and in particular required solving a large
SDP polynomially many times. In more detail, as mentioned in our introduction (lines 53–57), prior work gave an
ellipsoid-based method whose separation oracle is an SDP. As a result, this prior method is extremely impractical (and,
unsurprisingly, has not been implemented). One could also construct polynomial time algorithms for our problems that
iteratively filter outliers using an SDP in each iteration. (For sparse mean estimation, this algorithm is briefly described
in lines 151–157.) However, even this filter-based approach is quite slow, and in particular is very different than the
algorithms we design. As pointed out by the first reviewer, we are able to use the problem structure to eliminate the
need for *any* SDP. Ours are the first potentially practical robust algorithms for the problems considered.

**Reviewer 1:** We thank the reviewer for their careful reading of the paper and their positive feedback.

**Reviewer 2:** *Runtime and Comparison:* In the revised version of our paper we will include a plot of the running time.
For reference, we give a few numbers for robust sparse mean estimation here: (a) For $k = 1$, $d = 10$, $m = 50$: 0.005
seconds; (b) For $k = 10$, $d = 300$, $m = 50$: 0.014 seconds; (c) For $k = 40$, $d = 1000$, $m = 8000$: 1.4 seconds.

We would like to be able to directly compare to the prior published work on robust sparse mean estimation, but that
algorithm is quite complicated and has never been implemented. It also will surely be much worse: it uses the ellipsoid
algorithm and requires a large SDP as a separation oracle. For the same $k = 10$, $d = 300$, $m = 50$ case that our
algorithm solves in $0.014$ seconds, the very first SDP takes 10 seconds to solve with CVXOPT; the full ellipsoid-based
algorithm, if implemented, would take many times that.

*Overview of our Algorithms.* Here we expand upon the intuition for our robust sparse mean algorithm, in particular
lines 167–170. The goal is to produce a filter if the Frobenius norm of the difference between the empirical covariance
($\tilde{\Sigma}$) and the true covariance ($I$) is large on the largest $k^2$ entries — otherwise, if the difference is small, we don't need to
filter at all.

We use the terminology "good samples" (as in previous works in the area) to mean a set of samples satisfying certain
deterministic conditions that an uncorrupted Gaussian dataset will satisfy with high probability for a large enough
sample size. Our algorithms succeed under these deterministic conditions, which are described in the supplementary
material (e.g., Definition A.2 for the sparse mean case). One such condition is that the empirical expectation of every
degree-2 homogeneous polynomial $p(x)$ with $k^2$ nonzero coefficients is close to its true value. If $\|(\tilde{\Sigma} - I)_U\|_F$ is
too large, then we show that there exists such a polynomial $p$ that takes a large value in expectation, and hence on a
reasonable fraction of the sample points. But $p(x)$ is not large on average over a good set, so most of the points $x$ with
large $p(x)$ must be outliers. Therefore, we can remove the points with large $p(x)$ to filter out a set of mostly corrupted
points. (Notation clarification: As defined in the Appendix, $h_k(\cdot)$ is the thresholding operator which zeros out all but
the $k$ largest-magnitude entries of a vector.)

*Comments on Related Work:* The Cheng et al. [CDG19] robust mean estimation algorithm works for the dense case. In
particular, it has sample complexity $N = \tilde{\Omega}(d/\epsilon^2)$. That algorithm has no implications for the $k$-sparse setting studied
here, where we are interested in algorithms with sample complexity $N = \text{poly}(k, \log d)/\epsilon^2$.

While recent literature has developed robust mean estimation algorithms under more general distribution families, this is
*not* the case for the sparse setting studied here. The only previous algorithm for the sparse setting is the ellipsoid-based
method from [BDLS17] working under the same Gaussian assumptions as ours.

**Reviewer 3:** *Sample Complexity optimality:* The claim we make in lines 72–73 is that our $\tilde{O}(k^2 \log d/\epsilon^2)$ sample
complexity matches existing Statistical Query lower bounds, which hold for $k < \tilde{\Omega}(\sqrt{d})$. As the reviewer notes, above
this threshold, dense mean estimation algorithm performs better (and matches the SQ lower bound).

*Novelty of Our Filtering Algorithm:* The idea of filtering out outliers is of course not new. The question is how to find a
filter that removes the outliers. This is a problem-specific task that can be highly non-trivial.

As the reviewer seems to be suggesting, there is a similarity between ellipsoid-based methods and filtering methods.
But the existing ellipsoid-based method for robust sparse mean estimation relies on an SDP for its separation oracle.
The key contribution of our paper is to avoid convex programming entirely, producing a faster filter. To get such a filter
for robust sparse mean estimation, we actually need two filters: one linear and one *quadratic*. The quadratic filter has no
analog in prior work, yet (as we demonstrate in our experiments) is *necessary* for our approach. Analogous comments
apply for the robust sparse PCA setting.

[Meta-Review · NeurIPS 2019]

Congratulations! The reviewers all appreciated your paper and recommended its acceptance.